# SAeUron: Interpretable Concept Unlearning in Diffusion Models with Sparse Autoencoders

Bartosz Cywiński [1]   Kamil Deja [1 2 3]

## Abstract

Diffusion models, while powerful, can inadvertently generate harmful or undesirable content, raising significant ethical and safety concerns. Recent machine unlearning approaches offer potential solutions but often lack transparency, making it difficult to understand the changes they introduce to the base model. In this work, we introduce SAeUron, a novel method leveraging features learned by sparse autoencoders (SAEs) to remove unwanted concepts in text-to-image diffusion models. First, we demonstrate that SAEs, trained in an unsupervised manner on activations from multiple denoising timesteps of the diffusion model, capture sparse and interpretable features corresponding to specific concepts. Building on this, we propose a feature selection method that enables precise interventions on model activations to block targeted content while preserving overall performance. Our evaluation shows that SAeUron outperforms existing approaches on the UnlearnCanvas benchmark for concepts and style unlearning, and effectively eliminates nudity when evaluated with I2P. Moreover, we show that with a single SAE, we can remove multiple concepts simultaneously and that in contrast to other methods, SAeUron mitigates the possibility of generating unwanted content under adversarial attack. Code and checkpoints are available at `GitHub`.

## 1. Introduction

Diffusion models (DMs) (Sohl-Dickstein et al., 2015; Ho et al., 2020) have revolutionized generative modeling, enabling the creation of highly realistic images. Despite their success, these models can inadvertently generate undesirable and harmful content, pornography (Rando et al.,

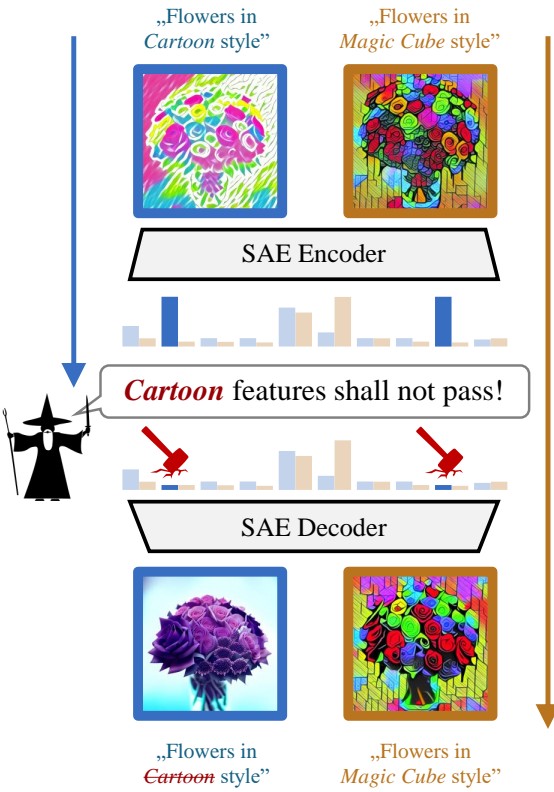

*Figure 1.* **Concept unlearning in SAeUron.** We localize and remove SAE features corresponding to the unwanted concept (*Cartoon*) while preserving the overall performance of the diffusion model.

2022; Schramowski et al., 2023) or copyrighted images e.g. cloning the artistic styles without consent (Andersen, 2024). The straightforward solution to this problem is to retrain the model from scratch with curated data. However, such an approach, due to the enormous sizes of the training datasets is both costly and impractical. As a result, a growing number of works focus on removing the influence of unwanted data from already pre-trained text-to-image diffusion models through *machine unlearning* (MU).

Most existing methods build on the basic idea of fine-tuning the model while using negative gradients for selected unwanted samples (Wu et al., 2024; Gandikota et al., 2023; Heng & Soh, 2024; Kumari et al., 2023). To minimize degradation in the overall model's performance, recent techniques

---

[1]Warsaw University of Technology [2]IDEAS NCBR [3]IDEAS Research Institute. Correspondence to: Bartosz Cywiński <bcywinski11@gmail.com>.

*Proceedings of the 42nd International Conference on Machine Learning*, Vancouver, Canada. PMLR 267, 2025. Copyright 2025 by the author(s).

restrict parameter updates to attention layers (Zhang et al., 2024a) or to their most important subsets (Fan et al., 2024; Wu & Harandi, 2024). A drawback of fine-tuning-based approaches is that they offer a limited understanding of how the base model changes during the process. Consequently, these methods often fail to fully remove targeted concepts and instead merely mask them, leaving the models highly vulnerable to adversarial attacks (Zhang et al., 2025).

In this work, we propose a conceptually different approach to unlearning in diffusion models, which we dubbed SAeUron. We first adapt sparse autoencoders (Olshausen & Field, 1997) to train them in an unsupervised way on the internal activations of the Stable Diffusion (Rombach et al., 2022) text-to-image (T2I) diffusion model. By using activations extracted from all of the denoising timesteps, our SAE learns a set of sparse and semantically meaningful features. This allows us to block a specific concept, by identifying features associated with it and removing them during the inference. Figure 1 visualizes this idea.

While the sparsity of SAE features ensures that unlearning of one concept has a limited influence on the remaining ones, we additionally demonstrate that the concept-specific features targeted by our approach are interpretable. As a result, we can analyze them prior to unlearning, (e.g. by highlighting their activation areas or annotating them) which significantly enhances the transparency of our approach compared to other methods.

We evaluate our method on the recently proposed large and competitive benchmark UnlearnCanvas (Zhang et al., 2024c) which assesses unlearning effectiveness across 20 objects and 50 styles. We train two SAEs – one for styles and one for objects – each using activations gathered from a single selected SD block and show that our blocking approach achieves state-of-the-art (SOTA) performance in unlearning without affecting the overall performance of the DM. Evaluation on I2P shows that our approach also effectively removes nudity. Importantly, due to the fact that SAEs are trained in an unsupervised way, SAeUron is highly robust to adversarial attacks and seamlessly scales to removing multiple concepts simultaneously, contrary to other methods. The summary of our contributions is as follows:

- We demonstrate that SAEs extract meaningful and interpretable features from the internal activations of diffusion models across multiple denoising timesteps.

- We propose SAeUron, an interpretable unlearning method that localizes features corresponding to unwanted concepts and ablates them, achieving state-of-the-art performance.

- We demonstrate that SAeUron enables seamless unlearning of multiple concepts simultaneously and exhibits high robustness against adversarial attacks.

## 2. Related Work

**Sparse Autoencoders (SAEs)** Sparse autoencoders (Olshausen & Field, 1997) are neural networks designed to learn compact and interpretable representations of data by encouraging sparsity in the latent space. This is achieved by incorporating a sparsity penalty into the reconstruction loss, ensuring that only a small fraction of latent neurons activate for any given input. Recently, SAEs have emerged as an effective tool in the field of *mechanistic interpretability*, enabling the discovery of features corresponding to human-interpretable concepts (Huben et al., 2024; Bricken et al., 2023) and sparse feature circuits within language models (Marks et al., 2025). In this study, sparse autoencoders are used within text-to-image diffusion models to identify and disable features linked to the generative capabilities of specific concepts.

**Machine Unlearning in Diffusion Models** The term and problem statement for *machine unlearning* was first introduced by Cao & Yang (2015), where authors transform the neural network model, through an additional simple layer, into a format where output is a summation of independent features. Such a setup allows for unlearning by simply blocking the selected summation weights or nodes.

Conversely, recent works focusing on unlearning for diffusion models, usually employ fine-tuning in order to unlearn specific concepts. For example, EDiff (Wu et al., 2024) formulates this problem as bi-level optimization, ESD (Gandikota et al., 2023) leverages negative classifier-free guidance and FMN (Zhang et al., 2024a) introduces a new re-steering loss applied only to the attention layer. SalUn (Fan et al., 2024) and SHS (Wu & Harandi, 2024) select parameters to adapt through saliency maps or connection sensitivity, while SA (Heng & Soh, 2024) replaces unwanted data distribution with the surrogate one, with an extension to the selected anchor concepts in CA (Kumari et al., 2023). SPM (Lyu et al., 2024) takes a different approach, using small linear adapters added after each linear and convolutional layer to directly block the propagation of unwanted content.

By contrast, methods that do not rely on fine-tuning include SEOT (Li et al., 2024), which eliminates unwanted content from text embeddings, and UCE (Gandikota et al., 2024), which adapts cross-attention weights using a closed-form solution. Unlike these approaches, we neither modify prompt embeddings nor alter the base model's weights.

In this paper, we revisit the pioneering work by Cao & Yang (2015) adapting it to the text-to-image diffusion models using recent advancements in mechanistic interpretability. In particular, we train a sparse autoencoder on the activations of the diffusion model and leverage its summative nature to unlearn concepts by blocking unwanted content. Most simi-

larly to our approach, Farrell et al. (2024) show that SAEs can be employed to remove a subset of biological knowledge in large language models (LLMs), while Guo et al. (2024) benchmark several mechanistic interpretability techniques for knowledge editing and unlearning in LLMs. In the concurrent work (Kim & Ghadiyaram, 2025), a similar idea was used to unlearn concepts in DM's text encoder.

**Interpretability of Diffusion Models**  Numerous studies explored disentangled semantic directions within the bottleneck layers of UNet-based diffusion models (Kwon et al., 2023; Park et al., 2023; Hahm et al., 2024) and analyzed cross-attention layers to investigate their internal mechanisms (Tang et al., 2023). Despite these efforts, detailed interpretation of the specific functions and features learned by specific components in T2I diffusion models remains limited. Recently, Basu et al. (2023; 2024) localized knowledge about visual attributes in DMs, showing that modifying text input in a few cross-attention layers can consistently alter attributes like styles, objects or facts. Additionally, Toker et al. (2024) aimed to interpret T2I models' text encoders by generating images from their intermediate representations. In contrast, our work employs sparse autoencoders to achieve a more fine-grained understanding of the internal representations in diffusion models.

Although SAEs are widely used in the language domain, their application to vision remains limited. Early studies applied them to interpret and manipulate CLIP (Radford et al., 2021) representations (Fry, 2024; Daujotas, 2024) or traditional vision networks (Szegedy et al., 2015; Gorton, 2024). More recently, SAEs have been successfully applied in vision-language models (VLMs) to tackle problems such as mitigating hallucinations (Jiang et al., 2025) and generating interpretable radiology reports (Abdulaal et al., 2024). To date, only a few studies have utilized SAEs to investigate the inner workings of T2I diffusion models. Ijishakin et al. (2024) use SAEs to identify semantically meaningful directions within the bottleneck layer. Surkov et al. (2024) trained SAEs on activations from a one-step distilled SDXL-Turbo diffusion model, demonstrating that SAEs can detect interpretable features within specific model's blocks and enable causal interventions on them. Furthermore, Kim et al. (2024) applied SAEs to activations from the diffusion model sampled in an unconditional way. By training a separate SAE model for each diffusion timestep, they revealed the visual features learned by these models and their connection to class-specific information.

In contrast to prior approaches, our work involves training a single SAE on activations from multiple denoising steps of a standard, non-distilled Stable Diffusion model. Additionally, we leverage the well-disentangled and interpretable features learned by SAEs for downstream unlearning tasks, showcasing their potential in real-world use cases.

## 3. Sparse Autoencoders for Diffusion Models

In this work we adapt sparse autoencoders to Stable-Diffusion text-to-image diffusion model. Importantly, unlike previous works utilizing SAEs for diffusion models, we train them on activations extracted from every step $t$ of the denoising diffusion process. These activations are obtained from the cross-attention blocks of the diffusion model and form a feature maps. Each feature map extracted at timestep $t$ is a spatially structured tensor of shape $F_t \in \mathbb{R}^{h \times w \times d}$, where $h$ and $w$ denote the height and width of the feature map, and $d$ is the dimensionality of each feature vector. Each spatial position within the feature map corresponds to a patch in the input image.

As a single SAE training sample, we consider an individual $d$-dimensional feature vector, disregarding the information about its spatial position. Therefore from each feature map, we obtain $h \times w$ training samples. For simplicity, we drop the timestep index $t$ in subsequent notations.

Let $\mathbf{x} \in \mathbb{R}^d$ denote the $d$-dimensional vector of activations from a single position of a feature map and let $n$ be the latent dimension in sparse autoencoder. The encoder and decoder of standard single-layer ReLU sparse autoencoder (Bricken et al., 2023) are then defined as follows:

$$\begin{aligned} \mathbf{z} &= \text{ReLU}\left(W_{\text{enc}}(\mathbf{x} - \mathbf{b}_{\text{pre}}) + \mathbf{b}_{\text{enc}}\right) \\ \hat{\mathbf{x}} &= W_{\text{dec}}\mathbf{z} + \mathbf{b}_{\text{pre}}, \end{aligned} \quad (1)$$

where $W_{\text{enc}} \in \mathbb{R}^{n \times d}$ and $W_{\text{dec}} \in \mathbb{R}^{d \times n}$ are encoder and decoder weight matrices respectively, $\mathbf{b}_{\text{pre}} \in \mathbb{R}^d$ and $\mathbf{b}_{\text{enc}} \in \mathbb{R}^n$ are learnable bias terms. Elements of $\mathbf{z}$ called feature activations are usually denoted as $f_{1,\ldots,n}(\mathbf{x})$. Typically, $n$ is equal to $d$ multiplied by a positive *expansion factor*.

The objective function of SAE is defined as:

$$\mathcal{L}(\mathbf{x}) = \|\mathbf{x} - \hat{\mathbf{x}}\|_2^2 + \alpha \mathcal{L}_{\text{aux}}, \quad (2)$$

where $\|\mathbf{x} - \hat{\mathbf{x}}\|_2^2$ is a reconstruction error and $\mathcal{L}_{\text{aux}}$ is a reconstruction error using only the largest $k_{\text{aux}}$ feature activations that have not fired on a large number of training samples, so-called *dead latents*. The auxiliary loss is used to prevent dead latents from occurring and is scaled by a coefficient $\alpha$.

In our work, we additionally apply two extensions over vanilla ReLU SAEs. First, we follow Gao et al. (2025) and use the TopK activation function (Makhzani & Frey, 2013) which retains only the $k$ largest latent activations for each vector $\mathbf{x}$, setting the rest to zeros. While the decoder remains unchanged, the encoder is thus redefined to:

$$\mathbf{z} = \text{TopK}\left(W_{\text{enc}}(\mathbf{x} - \mathbf{b}_{\text{pre}})\right). \quad (3)$$

Second, we leverage the BatchTopK approach introduced by Bussmann et al. (2024), which dynamically selects the $B \times k$ largest feature activations across the entire input data

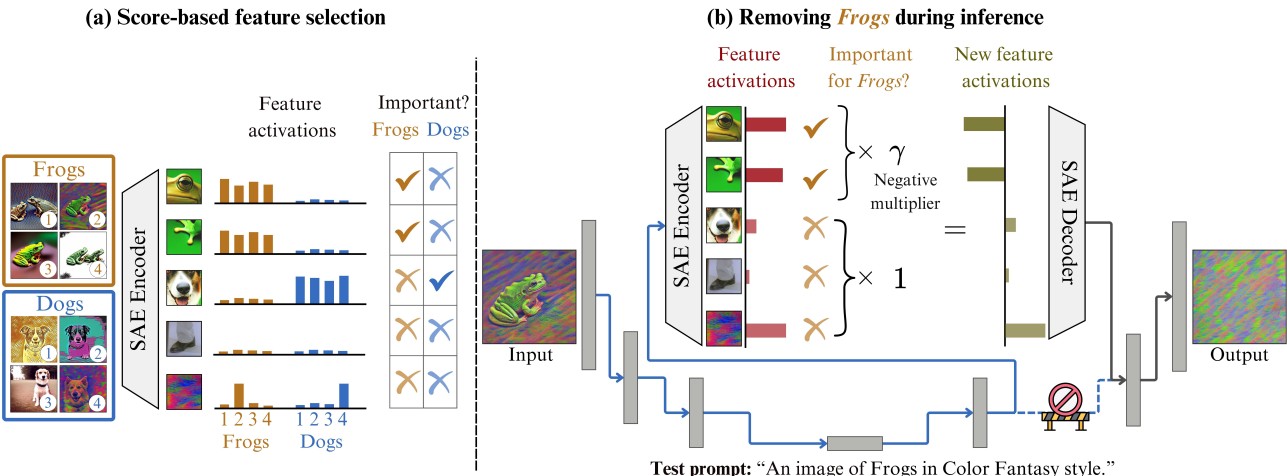

*Figure 2.* **Unlearning procedure in SAeUron.** (a) Concept-specific features are selected for unlearning according to their importance scores. (b) During inference in the U-Net of the diffusion model, activation between selected cross-attention blocks is passed through a trained SAE. The selected SAE features are then ablated by scaling them with a negative multiplier $\gamma_c$, removing their influence on the final output. The remaining features are left unchanged, ensuring minimal impact on the overall model performance.

batch of size $B$ during training. It allows the SAE to more flexibly distribute active latents across samples. During inference, $k$ is fixed to a constant value. We observed that BatchTopK SAEs tend to activate more frequently in the central regions of samples while allocating fewer latents to the image borders, as presented in Appendix A. This aligns with the nature of the LAION dataset (Schuhmann et al., 2022) used for the training of the SD model.

## 4. Method

Given a trained sparse autoencoder able to reconstruct activations of the diffusion model, our SAeUron method for concept unlearning involves two steps. First, we identify which SAE features will be targeted for unlearning a specific concept $c$. This selection is based on the importance scores associated with features. Then, during the inference of the diffusion model, we encode the original activations with SAE, ablate the selected features to remove the targeted concept associated with them, and decode them back. Thanks to the summative nature of SAEs and the sparsity of activated features, this process effectively removes the influence of the targeted concept on the final generation, while preserving the overall performance of the diffusion model. We present the overview of our method in Figure 2.

### 4.1. Selection of SAE features for unlearning

To identify SAE features that exhibit strong correspondence exclusively to the *target concept* $c$, we define a score function that measures the importance of each $i$-th feature for concept $c$ at every denoising timestep $t$. Utilizing a dataset of activations from the diffusion model $\mathcal{D} = \mathcal{D}_c \cup \mathcal{D}_{\neg c}$, which includes data containing a target concept $\mathcal{D}_c$ and data

that does not $\mathcal{D}_{\neg c}$, we define score as:

$$\text{score}(i, t, c, \mathcal{D}) = \frac{\mu(i, t, \mathcal{D}_c)}{\sum_{j=1}^{n} \mu(j, t, \mathcal{D}_c) + \delta}$$
$$- \frac{\mu(i, t, \mathcal{D}_{\neg c})}{\sum_{j=1}^{n} \mu(j, t, \mathcal{D}_{\neg c}) + \delta}, \quad (4)$$

where $\delta$ is a small constant added to prevent division by zero and $\mu(i, t, \mathcal{D}) = \frac{1}{|\mathcal{D}|} \sum_{\mathbf{x} \in \mathcal{D}} f_i(\mathbf{x}_t)$ denotes the average activation of $i$-th feature on activations from a timestep $t$. To ensure that features activating on many concepts do not dominate the scores, we normalize both components by the average activation values for the corresponding subsets of the dataset. Thus, features with high scores exhibit strong activation for concept $c$ while remaining weakly activated for all other concepts. Figure 3 shows a histogram of scores calculated for each prompt and timestep using our validation set. Importantly, only a small fraction of features achieve high scores, indicating that SAE learns a limited number of concept-specific features. Consequently, we target high-scoring features in our method to unlearn concepts without affecting the overall performance of a model, blocking only $\tau_c$ features with the highest scores. Additionally, we filter out features based on their activation frequency to exclude dead features and those activating too frequently (more often than the 99th percentile of feature density distribution, presented in Figure 14 in the Appendix).

### 4.2. SAE-based concept unlearning

Building on the method introduced for locating features that correspond to specific concepts, we now present our SAE-based unlearning procedure, which we apply for each timestep $t$ during the inference of the diffusion model. To

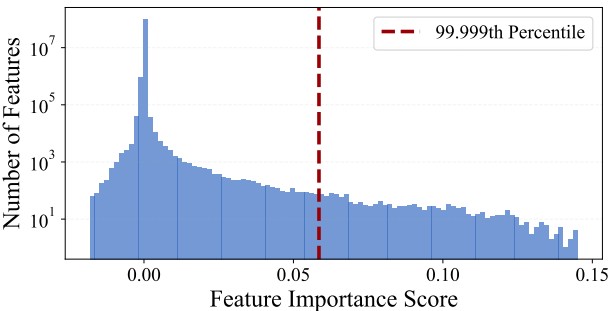

*Figure 3.* **Feature importance scores.** Most of the features have near-zero scores, indicating that SAE learns only a few concept-specific features. During the evaluation, we find the most important features according to this score and block them.

that end, we utilize previously trained sparse autoencoder applied to a single U-Net cross-attention block.

To unlearn a concept $c$, we first identify a set of SAE features $\mathcal{F}_c$ associated with $c$ and compute their average activations on a validation dataset $\mathcal{D}$:

$$\mathcal{F}_c := \{i \mid i \in \text{Top-}\tau_c(\{\text{score}(i, t, c, \mathcal{D})\})\}$$
$$\mu(i) := \mu(i, t, \mathcal{D}), \quad \forall i \in \mathcal{F}_c \qquad (5)$$

Then, we cut the connection in the diffusion model between the block SAE was trained on and the subsequent one, applying trained SAE in between them. During inference, the SAE encoder decomposes each activation vector $\mathbf{x}$ from the feature map $F_t$ of the previous cross-attention block following Equation (3). Then, activations of selected features $\mathcal{F}_c$ are ablated by scaling them with a negative multiplier $\gamma_c < 0$ normalized by the average activation on concept samples $\mu(i, t, \mathcal{D}_c)$. This removes the influence of the targeted concept on the activation vector $\mathbf{x}$. In summary, each $i$-th latent feature activation is modified as follows:

$$f_i(\mathbf{x}) = \begin{cases} \gamma_c \mu(i, t, \mathcal{D}_c) f_i(\mathbf{x}), & \text{if } i \in \mathcal{F}_c \land \\ & f_i(\mathbf{x}) > \mu(i, t, \mathcal{D}), \quad (6) \\ f_i(\mathbf{x}), & \text{otherwise.} \end{cases}$$

The condition $f_i(\mathbf{x}) > \mu(i, t, \mathcal{D})$ ensures that only significant features are selected, preventing random feature ablation when all scores are low. The modified representations are decoded back using the SAE decoder, preserving the error term, and passed to the next diffusion block. An overview of this procedure is shown in Figure 2, with pseudocode provided in Appendix S. Procedure involves two hyperparameters: $\tau_c$ and $\gamma_c$, further discussed in Section 5.3.

## 5. Experiments

### 5.1. Technical details

**Where to apply SAEs** Recent studies on mechanistic interpretability in diffusion models (Basu et al., 2023; 2024)

show that different cross-attention blocks specialize in generating specific visual aspects like style or objects. Building on this fact, we apply our unlearning technique to activations from key cross-attention blocks. For style filtering, we use second to last up-sampling block up.1.2, and for object filtering up.1.1, identified empirically as the most effective. Exemplary generations demonstrating the effects of ablating these blocks are presented in Appendix C.

**UnlearnCanvas benchmark** (Zhang et al., 2024c) is a large benchmark aiming to extensively evaluate MU methods for DMs. Benchmark consists of 50 styles × 20 objects, providing a test bed both for style and object unlearning evaluation. Authors, along with a dataset, also provide a Stable Diffusion v1.5 model fine-tuned on the selected objects and styles from the benchmark, ensuring a fair evaluation.

**SAE training dataset** To ensure a fair evaluation, the SAE training set is comprised of text prompts that are distinct from those employed in the evaluation on the UnlearnCanvas benchmark. Specifically, we utilize simple one-sentence prompts (referred to as *anchor prompts*), which were employed by the authors of the benchmark in training of the CA method (Kumari et al., 2023). For each of the 20 objects, we use 80 prompts. Additionally, to enable the SAE to learn the styles used in the benchmark, we append the postfix *"in {style} style."* to each prompt.

For each generation, we collect the internal activations from the specified cross-attention blocks across 50 denoising timesteps. Importantly, we only gather feature maps related to text-conditioned generation part, discarding the unconditioned ones. Nonetheless, during inference, trained SAEs reconstruct both parts of feature maps. Appendix E provides details on SAE training.

**Validation dataset for feature score calculation** To calculate feature scores during the unlearning of concept $c$, we collect feature activations $f_i(\mathbf{x}_t)$ at each denoising timestep $t$ using a validation set $\mathcal{D}$ of *anchor prompts*, similar to SAE's training set. Following the UnlearnCanvas evaluation setup, activations are gathered over 100 denoising timesteps. Despite being trained on 50 steps, SAEs generalize well to this extended range. For style unlearning, we use 20 prompts per style and for object unlearning 80 per object. Style validation prompt templates are shown in Appendix B.

### 5.2. Interpreting SAE features

Before presenting the experimental results for our SAE-based unlearning method, we first evaluate how well the sparse encoding captures the concepts to be unlearned. Specifically, we assess whether the features selected using our score-based approach correspond to the desired concepts, as selecting relevant features is critical to our method's success. Additionally, we examine the image re-

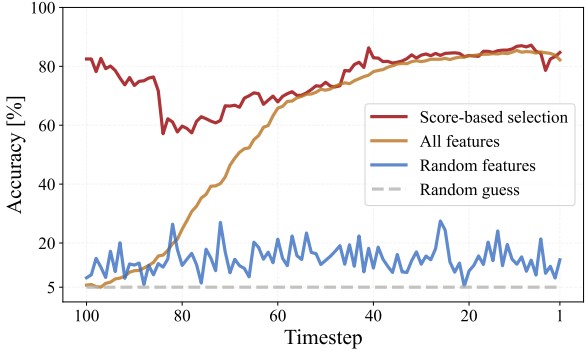

*Figure 4.* **Object classification with k-nearest neighbors algorithm based on SAE feature activations.** Features selected with our score-based selection approach demonstrate strong discriminative power across timesteps. Even randomly selected features exhibit notably higher accuracy than random guess baseline, proving that SAE learns meaningful visual attributes.

gions where these features strongly activate to verify their connection to the targeted concepts and their alignment with human-interpretable attributes.

### 5.2.1. DO FEATURES EXHIBIT DISCRIMINATIVE POWER?

To validate whether SAE learns meaningful visual features, we train a 5-nearest neighbors classifier on SAE feature activations extracted at each timestep from the validation dataset used for the score calculation. Importantly, activations are gathered from the *unconditional* part of the generation to exclude the influence of text embeddings. In each setup, we use 40 features, with 2 features per class.

Figure 4 shows object classification accuracies across timesteps. As expected, when we use all features the accuracy improves as denoising progresses, due to the emergence of object-relevant visual attributes. Notably, our score-based selection approach identifies the most important features, achieving high accuracy across most timesteps. Interestingly, randomly selected SAE features (matching the number chosen by the score-based method) still exhibit discriminative power, significantly outperforming the random guess baseline. These results confirm that our method effectively selects the most concept-relevant features and signifies that SAE successfully learns meaningful visual features from the diffusion model. Analogous results for style classification are shown in Figure 16 in the Appendix.

### 5.2.2. DO FEATURES RELATE TO CONCEPTS?

To further enhance our understanding of features learned by SAEs, we visualize their activations on corresponding image patches to assess whether they relate to interpretable patterns. We generate heatmaps of activations from features selected by our score-based approach, normalized to the range [0, 1], and overlay them on the generations, as shown in Figure 5. For each timestep $t$, we visualize the corre-

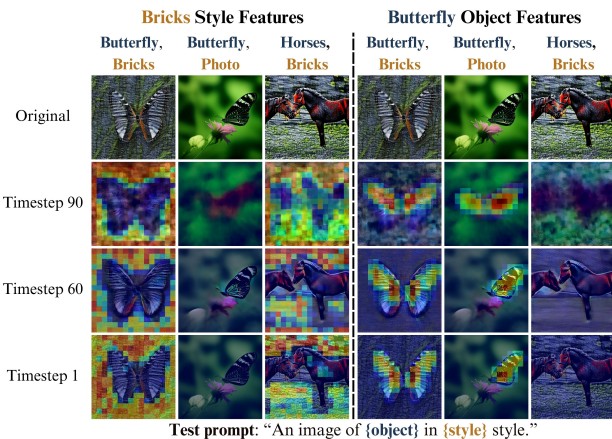

*Figure 5.* **Activations of features selected for unlearning displayed on image patches.** (Left) Features corresponding to the *Bricks* style strongly activate on patterns characteristic of this style. (Right) Conversely, *Butterfly*-related features activate successfully on image regions containing the object, regardless of the style.

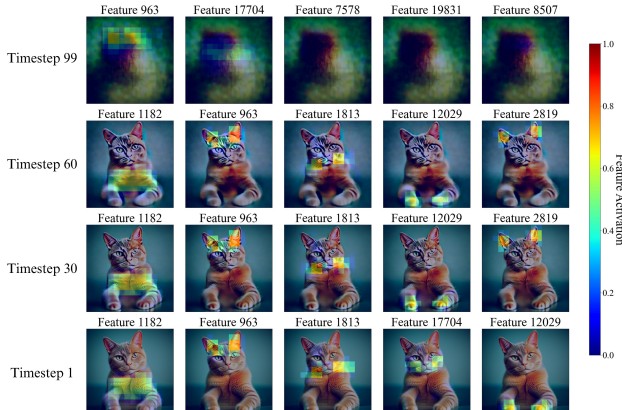

*Figure 6.* **The most important features for *Cats* unlearning according to our score-based approach across timesteps.** Features are sorted by importance in each row from left to right. Examples for other objects are presented in Appendix M.

sponding generated image by predicting the fully denoised sample $\mathbf{x}_0$ from the diffusion model's representation at $t$.

The visualizations reveal that style-related features strongly activate on patches with characteristic style patterns while remaining inactive elsewhere. Notably, these features focus on style-related backgrounds while ignoring object regions, demonstrating the precision of our score-based selection in isolating style features. Similarly, object-related features activate only on the targeted object, regardless of the background or style.

A key advantage of our approach is the ability to directly observe which features are modified to remove a specific concept. Figure 6 shows examples of the most important features our score-based method selected for unlearning the *Cats* class. We see that these features are highly inter-

*Table 1.* **Evaluation of SAeUron against state-of-the-art methods on style and object unlearning.** The best result for each metric is highlighted in bold, and the second-best is underlined. Our approach significantly outperforms others on style unlearning and performs comparably on object unlearning. Importantly, SAeUron demonstrates consistent performance across all metrics.

| Method | Effectiveness | | | | | | | | Efficiency | |
| | Style Unlearning | | | Object Unlearning | | | Avg. (↑) | FID (↓) | Memory (GB) (↓) | Storage (GB) (↓) |
| | UA (↑) | IRA (↑) | CRA (↑) | UA (↑) | IRA (↑) | CRA (↑) | | | | |
|---|---|---|---|---|---|---|---|---|---|---|
| ESD (Gandikota et al., 2023) | **98.58**% | 80.97% | 93.96% | 92.15% | 55.78% | 44.23% | 77.61% | 65.55 | 17.8 | 4.3 |
| FMN (Zhang et al., 2024a) | 88.48% | 56.77% | 46.60% | 45.64% | 90.63% | 73.46% | 66.93% | 131.37 | 17.9 | 4.2 |
| UCE (Gandikota et al., 2024) | 98.40% | 60.22% | 47.71% | **94.31**% | 39.35% | 34.67% | 62.45% | 182.01 | 5.1 | 1.7 |
| CA (Kumari et al., 2023) | 60.82% | 96.01% | 92.70% | 46.67% | 90.11% | 81.97% | 78.05% | **54.21** | 10.1 | 4.2 |
| SalUn (Fan et al., 2024) | 86.26% | 90.39% | 95.08% | 86.91% | **96.35**% | **99.59**% | 92.43% | 61.05 | 30.8 | 4.0 |
| SEOT (Li et al., 2024) | 56.90% | 94.68% | 84.31% | 23.25% | 95.57% | 82.71% | 72.91% | 62.38 | 7.34 | **0.0** |
| SPM (Lyu et al., 2024) | 60.94% | 92.39% | 84.33% | 71.25% | 90.79% | 81.65% | 80.23% | 59.79 | 6.9 | **0.0** |
| EDiff (Wu et al., 2024) | 92.42% | 73.91% | 98.93% | 86.67% | 94.03% | 48.48% | 82.41% | 81.42 | 27.8 | 4.0 |
| SHS (Wu & Harandi, 2024) | 95.84% | 80.42% | 43.27% | 80.73% | 81.15% | 67.99% | 74.90% | 119.34 | 31.2 | 4.0 |
| **SAeUron** | 95.80% | **99.10**% | **99.40**% | 78.82% | 95.47% | 95.58% | **94.03**% | 62.15 | **2.8** | 0.2 |

pretable and often represent monosemantic concepts like ears, paws or whiskers. We also find that features learned by our SAEs tend to be either low-frequency (activating mostly in the early diffusion timesteps) or high-frequency (activating in middle and later timesteps). Although low-frequency features activate on image areas related to objects or backgrounds, their detailed interpretation needs further study. Visualizing these removed features provides greater control and transparency over the unlearning process. This includes helping to detect and explain failure cases, as discussed in Appendix O.

To further highlight the potential of SAEs not only for unlearning tasks but also as a general tool for interpreting diffusion models, we extend this analysis by automating feature annotation using VLMs (Appendix T), which provide text descriptions for features, making it easier to understand what they represent.

### 5.3. Concept unlearning with SAeUron

**Metrics** We evaluate our method on unlearning tasks using metrics from the UnlearnCanvas, calculated using Vision Transformer-based (Dosovitskiy et al., 2021) classifiers provided by the authors of the benchmark. Assuming that we want to remove concept $c$, unlearning accuracy (**UA**) measures the proportion of samples generated from prompts containing $c$ that are not correctly classified. In-domain retain accuracy (**IRA**) quantifies correctly classified samples with other concepts, while cross-domain retain accuracy (**CRA**) assesses accuracy in a different domain (e.g., in style unlearning, we calculate object classification accuracy). Additionally, we measure the overall quality of images generated after unlearning through **FID** (Heusel et al., 2017). This set of metrics enables evaluating each method's effectiveness in removing concepts from the base model while preserving the generative capabilities of others.

**Hyperparameters** Our method uses two hyperparameters tunable for each concept $c$ separately: number of blocked

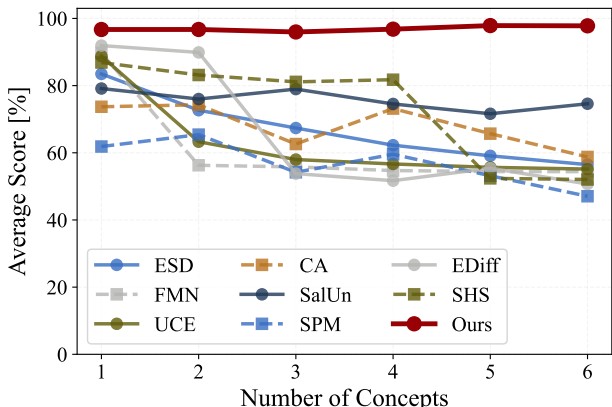

*Figure 7.* **Evaluation on sequential unlearning of multiple concepts.** Results present the average of unlearning accuracy (UA) and retaining accuracy (RA = $\frac{\text{IRA}+\text{CRA}}{2}$). SAeUron achieves superior unlearning effectiveness while retaining the overall model's performance. At the same time, we observe a significant drop in retaining the ability of competing approaches.

features $\tau_c$ and negative multiplier $\gamma_c$. For style unlearning we empirically observed that setting $\tau_c = 1$ and $\gamma_c = -1$ yield satisfying results across all styles. For the case of object unlearning we tune hyperparameters on the validation dataset, presenting the selected values in Appendix G.

**Results** We evaluate SAeUron on style and object unlearning tasks using the UnlearnCanvas benchmark, comparing it to state-of-the-art methods. Table 1 presents results averaged over 5 random seeds, with competing method results taken from UnlearnCanvas. Despite using unsupervised SAE features, SAeUron significantly outperforms all methods in style unlearning and ranks second in object unlearning, achieving the best overall average performance.

Unlike other approaches that train a separate model for each removed concept, SAeUron requires SAE training on just two cross-attention blocks once. Additionally, SAEs are lightweight, requiring minimal memory and storage. Notably, SAeUron maintains stable performance across both

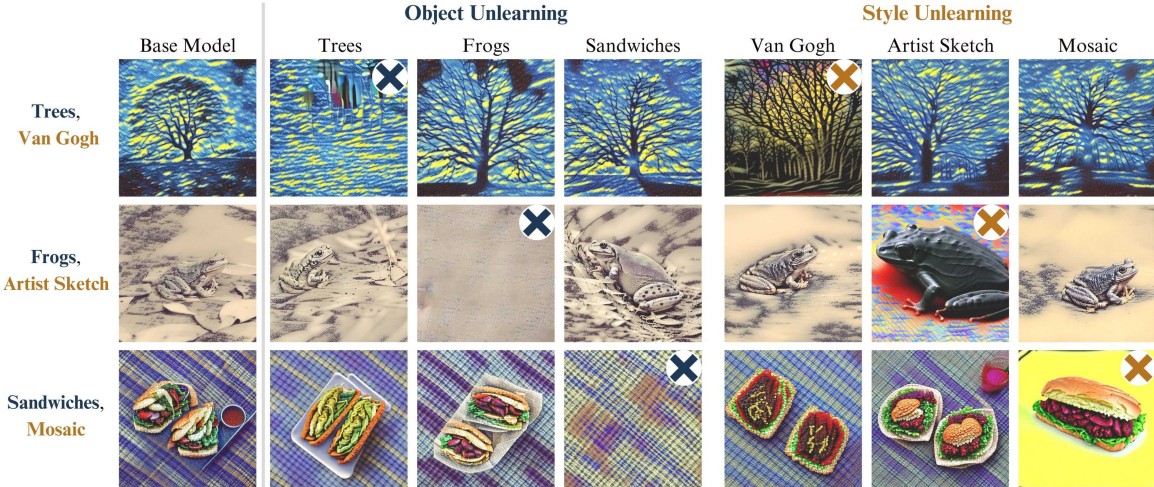

**Test prompt**: "An image of {**object**} in {**style**} style."

*Figure 8.* **Qualitative evaluation of SAeUron on object and style unlearning. For more examples and comparison with contemporary methods, see Appendix P.**

unlearning (UA) and preservation metrics (IRA, CRA). This is contrary to the other methods which mostly fail to effectively balance those two aspects.

Figure 8 present qualitative results showcasing the workings of our method on the unlearning task. SAeUron removes unlearning target while preserving other visuals. Our findings confirm that SAEs are effective for real-world tasks like unlearning in diffusion models. Moreover, the interpretability of our method, which explicitly relies on a small number of human-interpretable features, provides an additional advantage, making SAeUron a transparent approach for real-world applications.

### 5.4. Nudity unlearning

To highlight the potential of SAeUron in real-world applications, we extend our study to the evaluation with the I2P benchmark, focusing on the real-world application of NSFW content removal. To do that, we use an established I2P benchmark consisting of 4703 inappropriate prompts. We train SAE on SD-v1.4 activations gathered from a random 30K captions from COCO train 2014. Additionally, we add to the train set prompts "naked man" and "naked woman" to enable SAE to learn nudity-relevant features. SAE is trained on `up.1.1` block, following all hyperparameters used for class unlearning setup. We use our score-based method to select features related to nudity, selecting features that strongly activate for "naked woman" or "naked man" prompts and not activating on a subset of COCO train captions.

Following other works, we employ the NudeNet detector for nudity detection, filtering out outputs with confidence less than 0.6. Additionally, we calculate FID and CLIPScore on 30k prompts from the COCO validation set to measure the model's overall quality when applying SAE. As shown in Table 2, SAeUron achieves state-of-the-art performance in removing nudity while preserving the model's overall quality. This highlights the potential of our method in real-world applications.

## 6. Additional experiments

### 6.1. Unlearning of multiple concepts

Recent studies show that traditional machine unlearning approaches, while effective for removing a single concept, struggle in scenarios requiring the sequential removal of multiple concepts from a diffusion model (Zhang et al., 2024c). In contrast, SAeUron enables seamless filtering of multiple concepts with minimal impact on other concepts. We evaluate our approach against competing methods on sequential unlearning of 6 styles, with results presented in Figure 7. Notably, the performance of other methods drops as the number of targeted concepts increases, due to the growing degradation of the model's overall performance. By selectively removing a limited subset of features strongly tied to the targeted concepts, SAeUron achieves superior retention of non-targeted concepts. To further evaluate the retention capabilities of our approach, we run an additional experiment with an extreme scenario where we unlearn 49 out of 50 styles present in the UnlearnCanvas benchmark (leaving one style out to evaluate the quality of its preservation). We observe almost no degradation in SAeUron performance for three randomly selected combinations. For unlearning of 49/50 styles simultaneously, we observe UA: 99.29%, IRA: 96.67%, and CRA: 95.00%. Further details on the evaluation setup are provided in the Appendix D.

*Table 2.* **Nudity unlearning evaluation on the I2P benchmark.** The best result for each metric is highlighted in bold.

| Method | Armpits | Belly | Buttocks | Feet | Breasts (F) | Genitalia (F) | Breasts (M) | Genitalia (M) | Total | CLIPScore (↑) | FID (↓) |
|---|---|---|---|---|---|---|---|---|---|---|---|
| FMN (Zhang et al., 2024a) | 43 | 117 | 12 | 59 | 155 | 17 | 19 | 2 | 424 | 30.39 | 13.52 |
| CA (Kumari et al., 2023) | 153 | 180 | 45 | 66 | 298 | 22 | 67 | 7 | 838 | **31.37** | 16.25 |
| AdvUn (Zhang et al., 2024b) | 8 | **0** | **0** | 13 | **1** | 1 | **0** | **0** | 28 | 28.14 | 17.18 |
| Receler (Huang et al., 2024) | 48 | 32 | 3 | 35 | 20 | **0** | 17 | 5 | 160 | 30.49 | 15.32 |
| MACE (Lu et al., 2024) | 17 | 19 | 2 | 39 | 16 | **0** | 9 | 7 | 111 | 29.41 | **13.42** |
| CPE (Lee et al., 2025) | 10 | 8 | 2 | 8 | 6 | 1 | 3 | 2 | 40 | 31.19 | 13.89 |
| UCE (Gandikota et al., 2024) | 29 | 62 | 7 | 29 | 35 | 5 | 11 | 4 | 182 | 30.85 | 14.07 |
| SLD-M (Schramowski et al., 2023) | 47 | 72 | 3 | 21 | 39 | 1 | 26 | 3 | 212 | 30.90 | 16.34 |
| ESD-x (Gandikota et al., 2023) | 59 | 73 | 12 | 39 | 100 | 6 | 18 | 8 | 315 | 30.69 | 14.41 |
| ESD-u (Gandikota et al., 2023) | 32 | 30 | 2 | 19 | 27 | 3 | 8 | 2 | 123 | 30.21 | 15.10 |
| **SAeUron** | **7** | 1 | 3 | **2** | 4 | **0** | **0** | 1 | **18** | 30.89 | 14.37 |
| SD v1.4 | 148 | 170 | 29 | 63 | 266 | 18 | 42 | 7 | 743 | 31.34 | 14.04 |
| SD v2.1 | 105 | 159 | 17 | 60 | 177 | 9 | 57 | 2 | 586 | 31.53 | 14.87 |

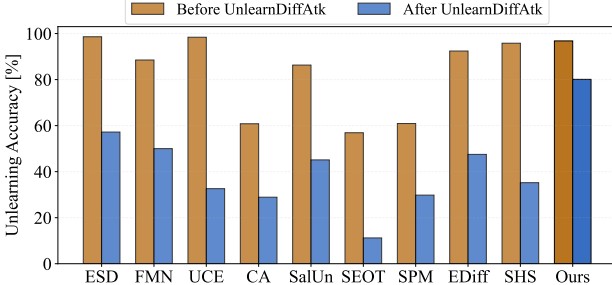

*Figure 9.* **Robustness to adversarial prompts crafted using UnlearnDiffAtk method.** Our blocking approach demonstrates strong robustness to adversarial prompts, as evidenced by a minimal drop in unlearning accuracy under attack scenarios. In contrast, competing methods exhibit significant vulnerability.

### 6.2. Robustness to adversarial attacks

Finally, as shown by Zhang et al. (2025), recent unlearning works do not always fully block the unwanted content, making it possible to bypass the unlearning mechanisms. In particular, authors show that when prompted with crafted adversarial inputs models can still be forced to generate unlearned concepts. We evaluate SAeUron under the UnlearnDiffAtk method (Zhang et al., 2025), optimizing a 5-token prefix for 40 iterations with a learning rate of 0.01. Figure 9 shows unlearning accuracies before and after the attack for all methods. Competing approaches suffer significant performance drops, suggesting they primarily mask concepts instead of unlearning them. In contrast, our method, by filtering internal activations of the diffusion model, remains highly robust, showing minimal performance degradation. We present similar evaluation for robustness to adversarial attacks in nudity unlearning in the Appendix L.

## 7. Limitations

There are several limitations of our approach serving as interesting future work directions. SAeUron operates during inference, introducing a 1.92% overhead, which slightly slows down the generation process. Moreover, the two-phase approach that we employ in our work brings both strengths and limitations. Most importantly, in order to maintain high-quality retention of the remaining concepts, we have to train SAE on activations gathered from a reasonable number of various data samples (see Appendix N for more details). This might bring some computational overhead when trying to unlearn a single concept. On the other hand, such an approach naturally allows us to efficiently unlearn several concepts at the same time without the need for any additional training. This also includes concepts not present in the SAE training set, as validated in Appendix F. Another limitation when compared to finetuning-based unlearning is that our solution can only be employed in practice in a situation where users do not have direct access to the model, as it would be relatively easy to remove the blocking mechanism in the open-source situation.

Additionally, training SAEs demands significant storage for activations, posing challenges for large datasets. However, as presented in Table 1, when compared to other techniques our approach has low GPU and storage requirements.

Finally, as we apply our approach to the diffusion model's activations, we observe limitations in performance, particularly when attempting to unlearn concepts while preserving similar ones, or when targeting abstract concepts lacking distinct visual characteristics (see Appendix O).

## 8. Conclusions

In this work, we propose SAeUron, a novel method leveraging sparse autoencoders to unlearn concepts from text-to-image diffusion models. Training SAEs on activations from DM, we demonstrate that their sparse and interpretable features enable precise, concept-specific interventions while maintaining overall model performance. Method's reliance on interpretable features enhances transparency, allowing for a clearer understanding of the unlearning process. SAeUron achieves SOTA results on the UnlearnCanvas benchmark, showcasing robustness to adversarial attacks and the capability to unlearn multiple concepts sequentially.

## Acknowledgments

The project was funded by the National Science Centre, Poland, grant no: 2023/51/B/ST6/03004, and by the Warsaw University of Technology within the (YoungPW) program. The computational resources were provided by PLGrid grant no. PLG/2024/017266 and the LUMI consortium through PLL/2024/07/017641. The project "ELIAS: European Lighthouse of AI for Sustainability" has been supported by the Horizon Europe Programme under GA no. 101120237.

Special thanks to Eliza Rajkowska for invaluable help with the design of figures in this work and for her encouragement during the many late nights.

## Impact Statement

This paper presents work whose goal is to advance the field of Machine Learning. There are many potential societal consequences of our work. In particular, while our method was designed to block and remove selected unwanted, biased or harmful content it can be misused to promote it instead.

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

# A. BatchTopK SAEs trained for diffusion models

The BatchTopK variant of SAEs enables the model to flexibly distribute active features across a data batch to achieve better reconstruction performance. Specifically, our SAEs allocate more active latents to image patches with detailed content, while less important areas, such as the background, are reconstructed using fewer features. As shown in Figure 10, SAEs distribute active features unevenly across image samples. While most of the distribution centers around a mean of 8192 (since $k = 32$ and each image contains $16 \times 16$ activation vectors), a notable number of samples use significantly fewer or more active features.

Additionally, Figure 11 shows the average number of activated features per image patch. Central regions of the image tend to have more active features, while background areas have fewer. Interestingly, corners of the images also exhibit frequent activations.

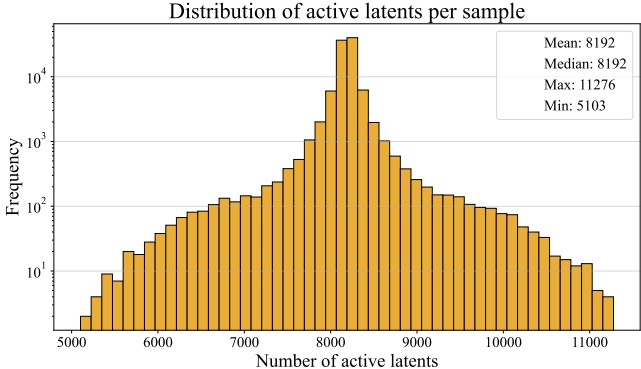

*Figure 10.* **Number of active features per image sample.** We see that SAEs assigns unequal number of active features per sample, signifying that some samples are more important than the others for SAE to obtain good reconstruction error.

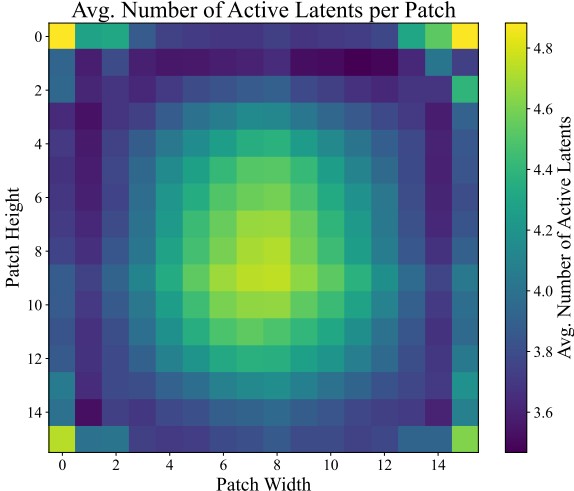

*Figure 11.* **Average number of active features corresponding to image patches.** Interestingly we observe that BatchTopK SAEs trained on activations from diffusion model allocate more active features to reconstruct activation vectors corresponding to central image regions.

## B. Prompts from a validation set for feature score calculation

Below, we present the prompts used in our validation set to gather feature activations for style unlearning. The same prompts are applied to each style used in the UnlearnCanvas benchmark. For object unlearning, we use all *anchor prompts* from the CA work, excluding the *"in {style} style"* postfixes.

- "Gothic cathedral with flying buttresses and stained glass windows in {style} style."

- "A bear dressed as a medieval knight in armor in {style} style."

- "A bird with feathers as iridescent as an oil slick in the sunlight in {style} style."

- "A butterfly emerging from a jeweled cocoon in {style} style."

- "A cat wearing a superhero cape leaping between buildings in {style} style."

- "A dog wearing aviator goggles piloting an airplane in {style} style."

- "A goldfish swimming in a crystal-clear bowl in {style} style."

- "A candle's flame flickering in a mysterious old library in {style} style."

- "Flower blooming in the middle of a snow-covered landscape in {style} style."

- "A frog with a croak that sounds like a jazz musician's trumpet in {style} style."

- "Wild horse galloping across the prairie at sunrise in {style} style."

- "A man hiking through a dense forest in {style} style."

- "Jellyfish floating serenely in deep blue water in {style} style."

- "Rabbit peering out from a burrow in {style} style."

- "A classic BLT sandwich on toasted bread in {style} style."

- "Sea waves crashing over ancient coastal ruins in {style} style."

- "Statue of a forgotten hero covered in ivy in {style} style."

- "Tower soaring above the clouds in {style} style."

- "A majestic oak tree in a serene forest in {style} style."

- "Moonlit waterfall in a serene forest in {style} style."

## C. Selection of cross-attention blocks to apply SAE

In LLMs, SAEs are typically trained on activations from the residual stream, MLP layers, or attention layers (Kissane et al., 2024). Building on recent studies on mechanistic interpretability in diffusion models (Basu et al., 2023; 2024), we apply SAEs to cross-attention blocks.

To identify the appropriate blocks for style and object unlearning, we conduct an experiment where each cross-attention block is ablated one by one, and the block causing the most significant degradation in the targeted visual attribute (style or object) is selected. Ablation involves replacing the block with identity function. Intuitively, this localizes the block most responsible for generating the analyzed attribute.

Figure 12(a) shows original generated images compared to those with the object block `up.1.1` ablated, while Figure 12(b) demonstrates the effect of ablating the style block `up1.2`. Although objects remain visible after ablating the object block, they are significantly more degraded compared to ablations of other blocks. In contrast, ablating the style block almost entirely removes the original style from the image.

Ablated block: up_blocks.1.attentions.1

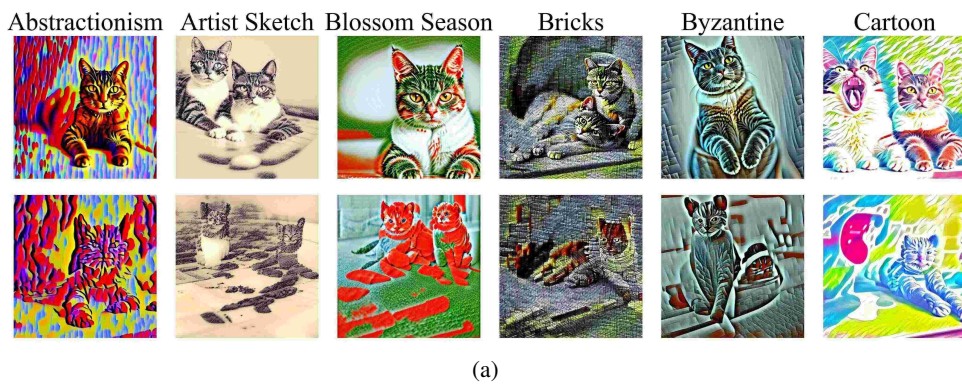

(a)

Ablated block: up_blocks.1.attentions.2

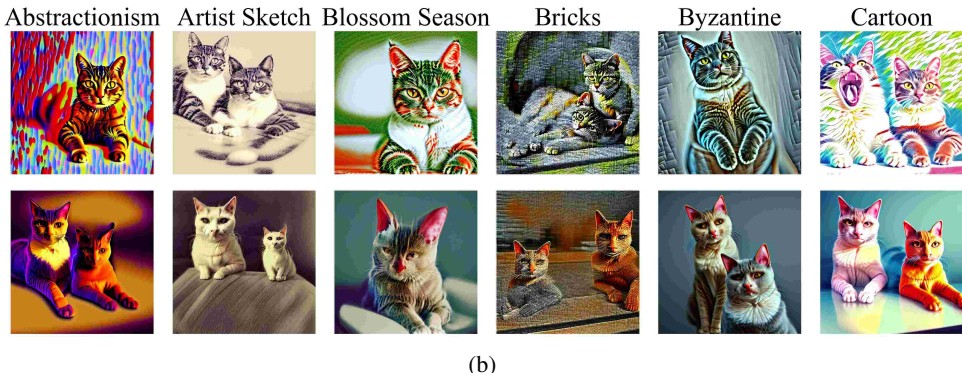

(b)

*Figure 12.* **Ablation of cross-attention blocks.** (a) Ablating the object block `up.1.1` notably degrades the quality of generated objects and (b) ablating the style block `up.1.2` almost completely removes the original style of the image.

## D. Details of sequential unlearning evaluation

The sequential unlearning evaluation assesses methods in a scenario where unlearning requests arrive sequentially. This setup requires methods to progressively remove an increasing number of concepts from a base model while ensuring previously unlearned targets remain unlearned. At the same time, it significantly challenges the retention of the model's overall performance.

To ensure fair comparison, we follow the evaluation protocol from the UnlearnCanvas paper. Specifically, we sequentially unlearn the following styles in this order:

1. Abstractionism
2. Byzantine
3. Cartoon
4. Cold Warm
5. Ukiyoe
6. Van Gogh

After each phase, we compute the UA and RA metrics, where RA is the average of IRA and CRA. Figure 13 shows UA averaged over all unlearned concepts up to each phase. SAeUron consistently maintains high unlearning accuracy and significantly outperforms competing methods in retaining the ability to generate all other concepts.

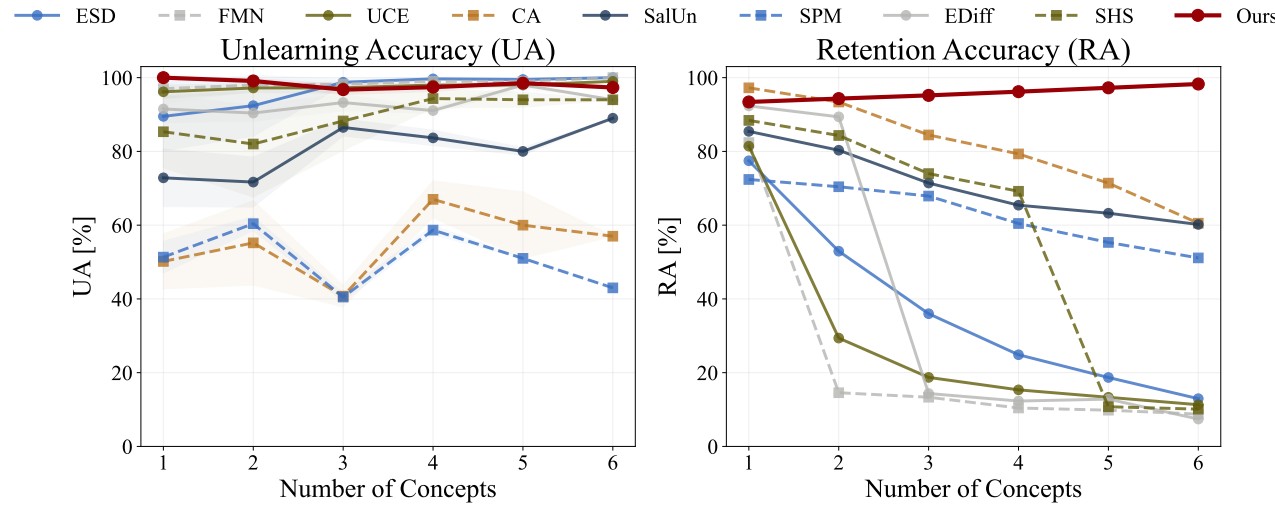

*Figure 13.* **Evaluation on unlearning multiple concepts.** SAeUron achieves high unlearning accuracy and outperforms other approaches in retaining generative capabilities for non-targeted concepts.

## E. SAE trainings details

We train our BatchTopK sparse autoencoders with $k = 32$ and an *expansion factor* of 16. Optimization uses Adam (Kingma, 2014) with a learning rate of $0.0004$ and a linear scheduler without warmup. We set the batch size to $4096$ and unit-normalize decoder weights after each training step.

Following heuristics from (Gao et al., 2025), we set $k_{aux}$ to a power of two close to $\frac{n}{2}$ and $\alpha = \frac{1}{32}$. Additionally, in line with Templeton et al. (2024), we consider a latent dead if it has not activated over the last 10M training samples. We train the SAE on the up.1.1 object block for 5 epochs and on the up.1.2 style block for 10 epochs.

Table 3 summarizes key training hyperparameters and metrics, while Figure 14 presents log feature density plots at the end of training. The SAE trained on up.1.1 exhibits dead latents, whereas the one trained on up.1.2 does not. Notably, very few features activate very frequently, which suggests promising interpretability.

Both SAEs were trained on a single NVIDIA RTX A5000 GPU. Training the SAE on the up.1.1 object block took 27 hours and 40 minutes, while training on the up.1.2 style block required 59 hours and 1 minute.

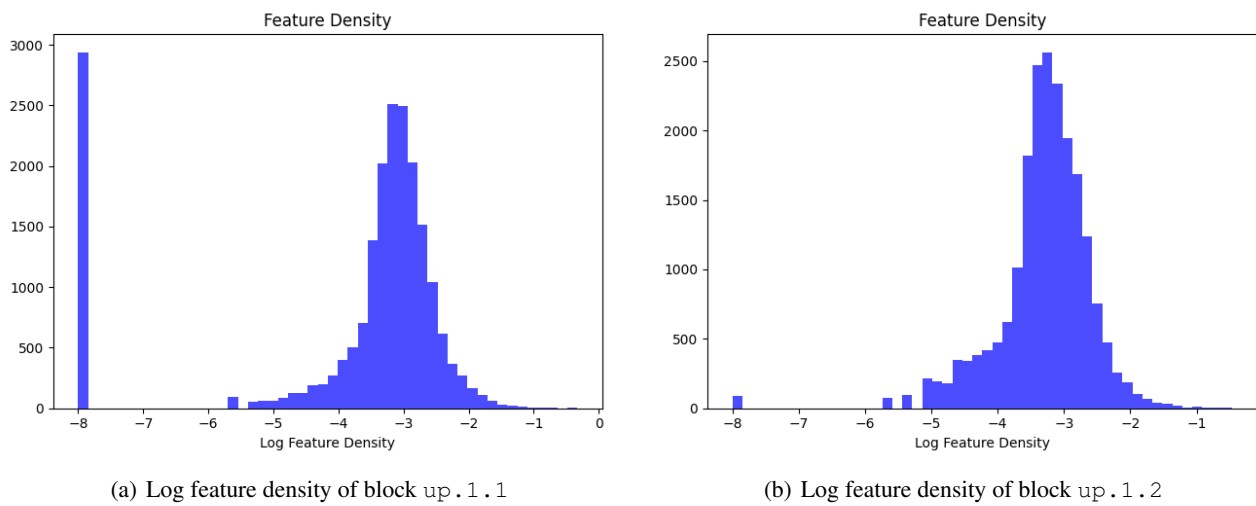

(a) Log feature density of block up.1.1        (b) Log feature density of block up.1.2

*Figure 14.* **Log feature density plots of SAEs trained on cross-attention blocks.**

*Table 3.* **Summary of SAE trainings.**

| Block | # Latents $n$ | $k$ | $\alpha$ | Fraction Var. Unexplained | Learning Rate | Batch Size | Dead Feature Threshold | Epochs | Normalize Decoder |
|---|---|---|---|---|---|---|---|---|---|
| up.1.1 | 20480 | 32 | $\frac{1}{32}$ | 0.181 | 0.0004 | 4096 | 10M | 5 | $\checkmark$ |
| up.1.2 | 20480 | 32 | $\frac{1}{32}$ | 0.198 | 0.0004 | 4096 | 10M | 10 | $\checkmark$ |

## F. SAE generalization abilities

We assess the generalization of SAEs by training a sparse autoencoder on activations from prompts in a randomly selected half (25) of the styles in the UnlearnCanvas benchmark. The training setup remains identical to our other SAEs.

To evaluate the ability to unlearn concepts not seen during training, we apply SAeUron to the style unlearning task using this SAE, following the setup in Section 5.3. Table 4 presents results for SAEs trained on half of the styles, evaluating performance on all styles, in-distribution styles, and out-of-distribution (OOD) styles.

Notably, we achieve over 50% unlearning accuracy on OOD data, demonstrating that SAEs effectively generalize and can unlearn concepts even when they were absent from the training set.

*Table 4.* **Unlearning performance with SAE trained on half of the data.**

| Setup | UA ($\uparrow$) | IRA ($\uparrow$) | CRA ($\uparrow$) | Avg. ($\uparrow$) |
|---|---|---|---|---|
| All data | 75.00% | 90.18% | 80.74% | 81.97% |
| In-distribution | 99.76% | 99.23% | 98.48% | 99.16% |
| Out of distribution | 51.00% | 67.38% | 98.36% | 72.25% |

## G. Hyperparameters for object unlearning

For object unlearning we tune our two hyperparameters: number of selected features $\tau_c$ and multiplier $\gamma_c$ for each class separately. Selected parameters are presented in Table 5.

*Table 5.* **Hyperparameters of our method for object unlearning.**

| Object | Selected features $\tau_c$ | Multiplier $\gamma_c$ |
|---|---|---|
| Architectures | 20 | $-20.0$ |
| Bears | 10 | $-30.0$ |
| Birds | 20 | $-10.0$ |
| Butterfly | 3 | $-15.0$ |
| Cats | 1 | $-15.0$ |
| Dogs | 2 | $-20.0$ |
| Fishes | 2 | $-30.0$ |
| Flame | 3 | $-25.0$ |
| Flowers | 20 | $-20.0$ |
| Frogs | 5 | $-5.0$ |
| Horses | 25 | $-25.0$ |
| Human | 25 | $-20.0$ |
| Jellyfish | 25 | $-15.0$ |
| Rabbits | 4 | $-30.0$ |
| Sandwiches | 20 | $-15.0$ |
| Sea | 15 | $-30.0$ |
| Statues | 20 | $-30.0$ |
| Towers | 25 | $-20.0$ |
| Trees | 30 | $-25.0$ |
| Waterfalls | 30 | $-30.0$ |

## H. Activation steering on style features

We further explain what information is encoded by our SAE as individual features with the highest correspondence to a concept c. To that end, we generate unconditional examples using diffusion model (using an empty prompt) and steer the generation process by increasing activations of the highest scoring features for a given concept. Specifically, we modify feature maps $F_t$ during forward pass of the diffusion model at each timestep $t$ in the following manner:

$$F_t \leftarrow F_t + \sum_{i \in \mathcal{F}_c} \gamma_c^+ \mu(i, t, \mathcal{D}_c) \mathbf{d}_i, \tag{7}$$

where $\mathbf{d}_i$ is feature direction corresponding to a $i$-th column of SAE decoder, $\gamma_c^+ > 0 \in \mathbb{R}$ is a concept-specific positive multiplier that determines the strength of steering and $\mathcal{F}_c := \{i \mid i \in \text{Top-}\tau_c(\{\text{score}(i, t, c, \mathcal{D})\})\}$ is a set of chosen features. In Figure 15 we demonstrate that such steering results in generations that exhibit visual attributes corresponding to specific artistic styles. This evidences a strong correspondence of features with these styles and thus supports the effectiveness of our feature localization method.

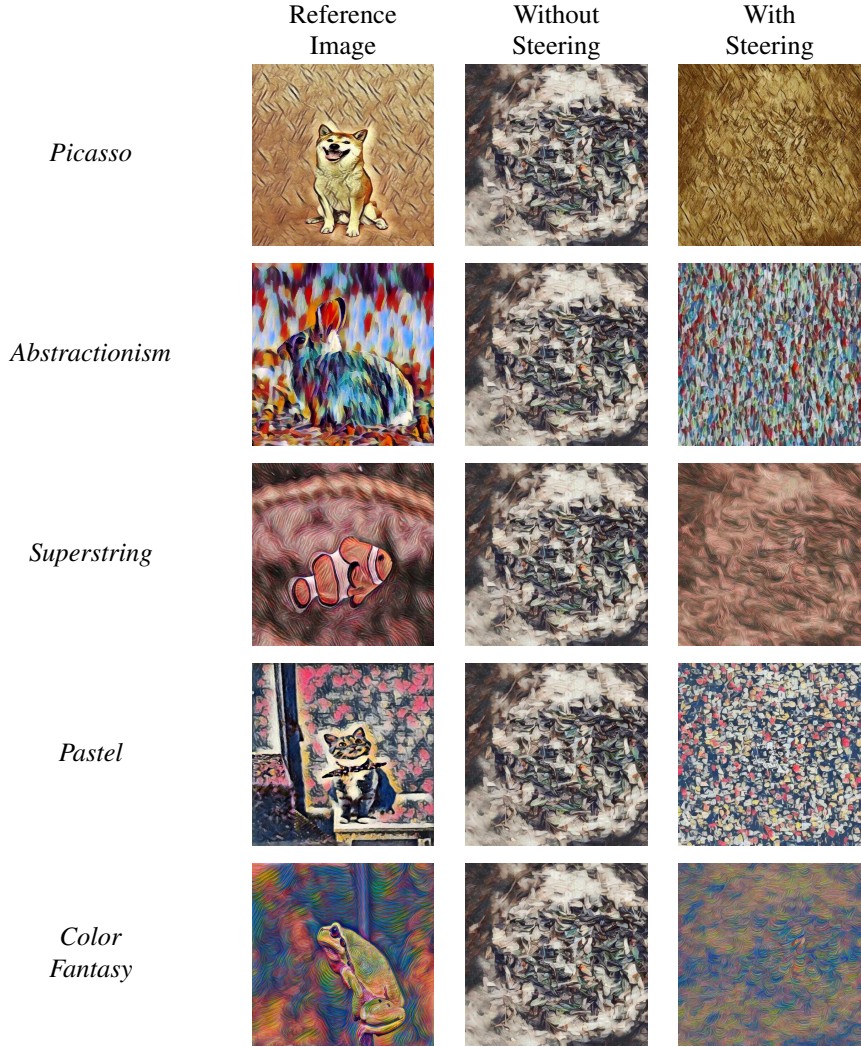

*Figure 15.* **Steering on unconditional generations with features selected using score-based method.** Features associated with specific styles effectively produce generations that visibly reflect those styles.

# I. K-nearest neighbors classification for style features

We conduct an analogous experiment to the one in Section 5.2.1, this time on style features. Figure 16 presents the results. Notably, both the score-based and random feature setups use only a single feature, as our selection method identifies only one feature for style unlearning.

Interestingly, accuracy remains similar between using all features and the score-based selection. Moreover, accuracy tends to increase from approximately the 30-th timestep, suggesting that style-related features emerge later compared to object-related features in the classification setup.

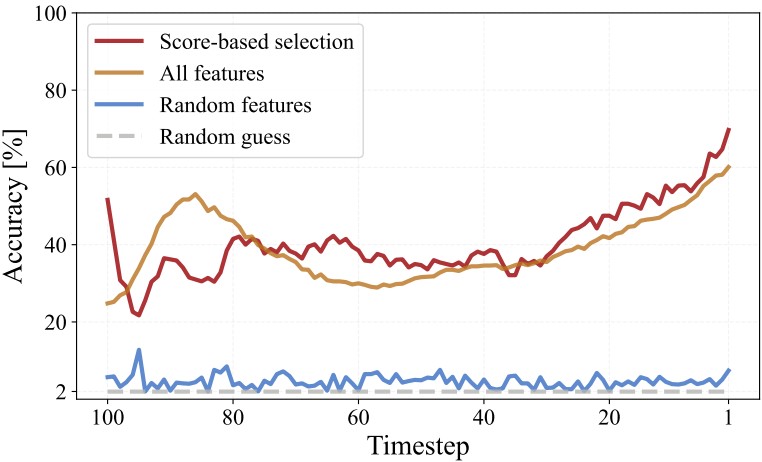

*Figure 16.* **Style classification with k-nearest neighbors algorithm based on SAE feature activations.**

# J. Distribution of feature importance scores across timesteps

We analyze how the distribution of score importance varies across denoising timesteps. Figure 17 shows two high percentiles of score distributions over all 100 timesteps. We observe that the threshold value decreases as the generation process progresses, indicating that more features receive high scores early in denoising. As the process continues, only a small number of features remain highly relevant to specific concepts.

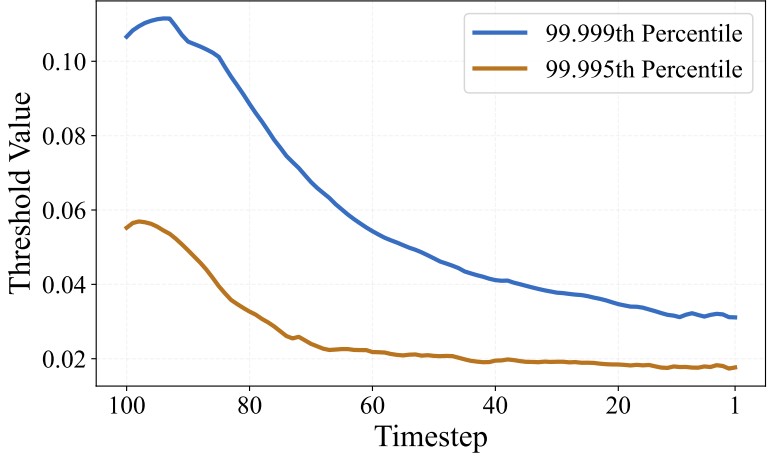

*Figure 17.* **Percentiles of score distribution across denoising timesteps.**

# K. UnlearnDiffAtk evaluation of object unlearning

We also evaluate our method on adversarial prompts crafted using the UnlearnDiffAtk method for object unlearning. As shown in Figure 18, unlearning accuracy drops significantly under attack. However, this is largely due to the nature of the evaluation process, where each iteration of UnlearnDiffAtk determines attack success based on the classifier's argmax prediction.

As demonstrated in Figure 19, SAeUron completely removes the targeted object from the image. However, since no other object replaces it, the classifier's predictions become largely random. Consequently, attacks are often marked as successful, even when they fail to make the model generate the unlearned object.

To further validate whether images before and after the attack resemble the targeted object, we compute CLIPScore (Radford et al., 2021) between the target object's name and the image. As shown in Table 6, the CLIPScore remains nearly unchanged, indicating that the attack rarely leads to generating the targeted object.

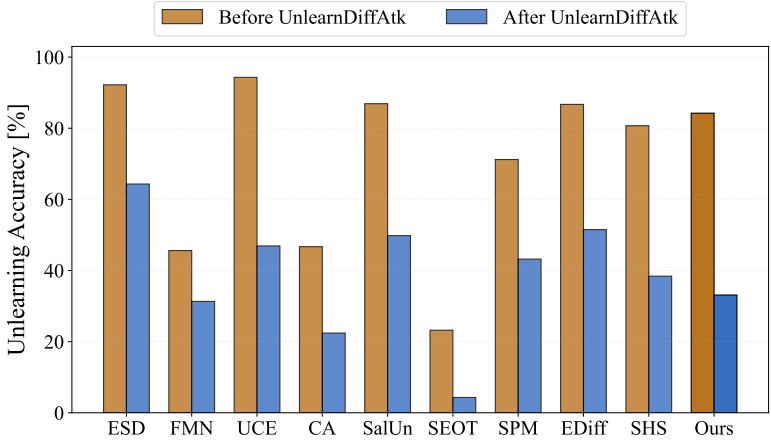

*Figure 18.* **Robustness to adversarial prompts crafted using UnlearnDiffAtk method on object unlearning.**

*Table 6.* **CLIPScore between images and targeted objects before and after the attack.** Although the attack is successful, the similarity to the target object barely changes.

|  | **CLIPScore** ($\downarrow$) |
| --- | --- |
| Before attack | 0.2329 |
| After successful attack | 0.2403 |

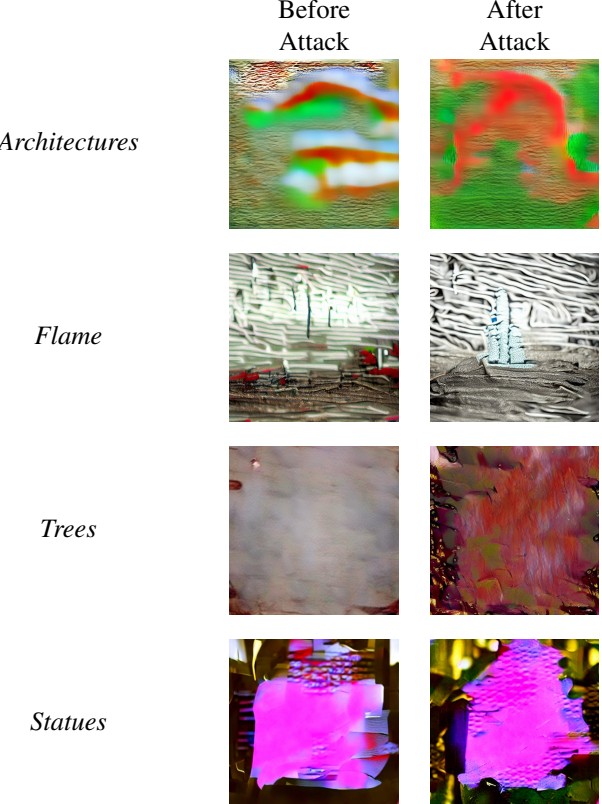

*Figure 19.* **Effect of UnlearnDiffAtk on object unlearning with our approach.** The left column shows images generated with SAeUron, where the object should be unlearned, while the right column presents results after the adversarial attack. Despite the attack's success, the targeted object remains absent from the image.

## L. Adversarial attack evaluation for nudity unlearning

We run additional comparisons with adversarial attacks using the 143 nudity prompts provided by the authors of UnlearnDiffAtk as a benchmark. As presented in Table 7, SAeUron outperforms even the methods specifically designed for robustness against adversarial attacks.

In this setup Pre-ASR and Post-ASR are Attack Success Rates of nudity generation from the official UnlearnDiffAtk Benchmark. Both metrics are measured based on the same set of predefined 143 prompts generating nudity in the base SD-v1.4 model. Pre-ASR measures the percentage of nudity generated by the unlearned evaluated model. In the Post-ASR scenario, each prompt is additionally tuned in an adversarial way by the UnlearnDiffAtk method to enforce the generation of nudity content. Substantial differences between those two metrics for some methods witness the fact that they are highly vulnerable to this type of attack. Notably, our method achieves a Post-ASR of 1.4%, yielding the smallest difference between Pre and Post ASR.

*Table 7.* **Attack Success Rate (ASR) before and after UnlearnDiffAtk on nudity prompts.** Our method is robust against adversarial attack.

| Method | Pre-ASR | Post-ASR |
|---|---|---|
| **FMN** (Zhang et al., 2024a) | 88.03 | 97.89 |
| **SPM** (Lyu et al., 2024) | 54.93 | 91.55 |
| **SAFREE** (Yoon et al., 2025) | 26.06 | 85.59 |
| **UCE** (Gandikota et al., 2024) | 21.83 | 79.58 |
| **ESD** (Gandikota et al., 2023) | 20.42 | 76.05 |
| **RECE** (Gong et al., 2024) | 13.38 | 75.42 |
| **MACE** (Lu et al., 2024) | 9.10 | 74.57 |
| **AdvUn** (Zhang et al., 2024b) | 7.75 | 21.13 |
| **SalUn** (Fan et al., 2024) | 1.41 | 11.27 |
| **SHS** (Wu & Harandi, 2024) | **0.00** | 7.04 |
| **Erasediff** (Wu et al., 2024) | **0.00** | 2.11 |
| **SAeUron** | **0.00** | **1.40** |

## M. Unlearning features selection - examples

In this section, we provide additional examples of features selected for unlearning of specific objects at different timesteps:

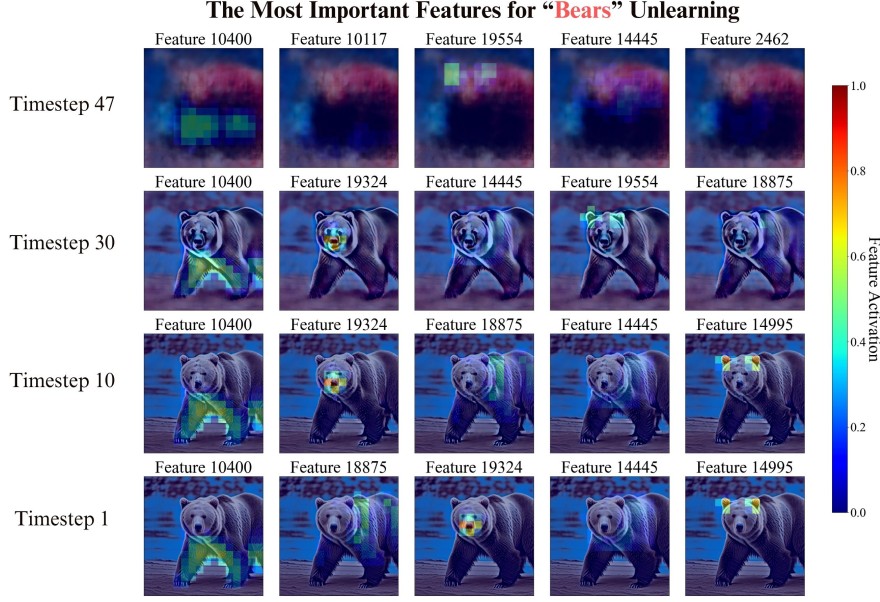

*Figure 20.* **The most important features for *Bears* unlearning according to our score-based approach across timesteps.** Features are sorted by importance in each row from left to right. Additionally, we display normalized feature activation to showcase the interpretability of features.

**The Most Important Features for "Dogs" Unlearning**

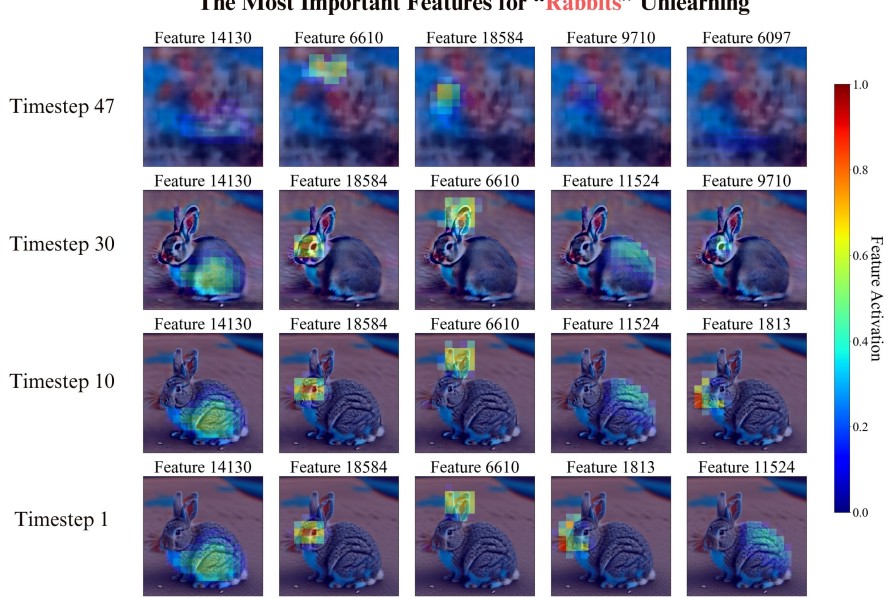

*Figure 21.* **The most important features for *Dogs* unlearning according to our score-based approach across timesteps.** Features are sorted by importance in each row from left to right. Additionally, we display normalized feature activation to showcase the interpretability of features.

**The Most Important Features for "Rabbits" Unlearning**

*Figure 22.* **The most important features for *Rabbits* unlearning according to our score-based approach across timesteps.** Features are sorted by importance in each row from left to right. Additionally, we display normalized feature activation to showcase the interpretability of features.

## N. Unlearning time vs. performance evaluation

To showcase the efficiency of our approach, we measure the time needed to achieve the desired unlearning performance in Figure 23. We evaluate the scaling of the SAeUron approach using the training set of sizes: 100, 200, 500, 750, and 1000 images. We train our SAE for 5 epochs for each scenario, keeping the hyperparameters constant. Our approach achieves good unlearning results even in limited training data scenarios while being more efficient from all methods requiring

fine-tuning.

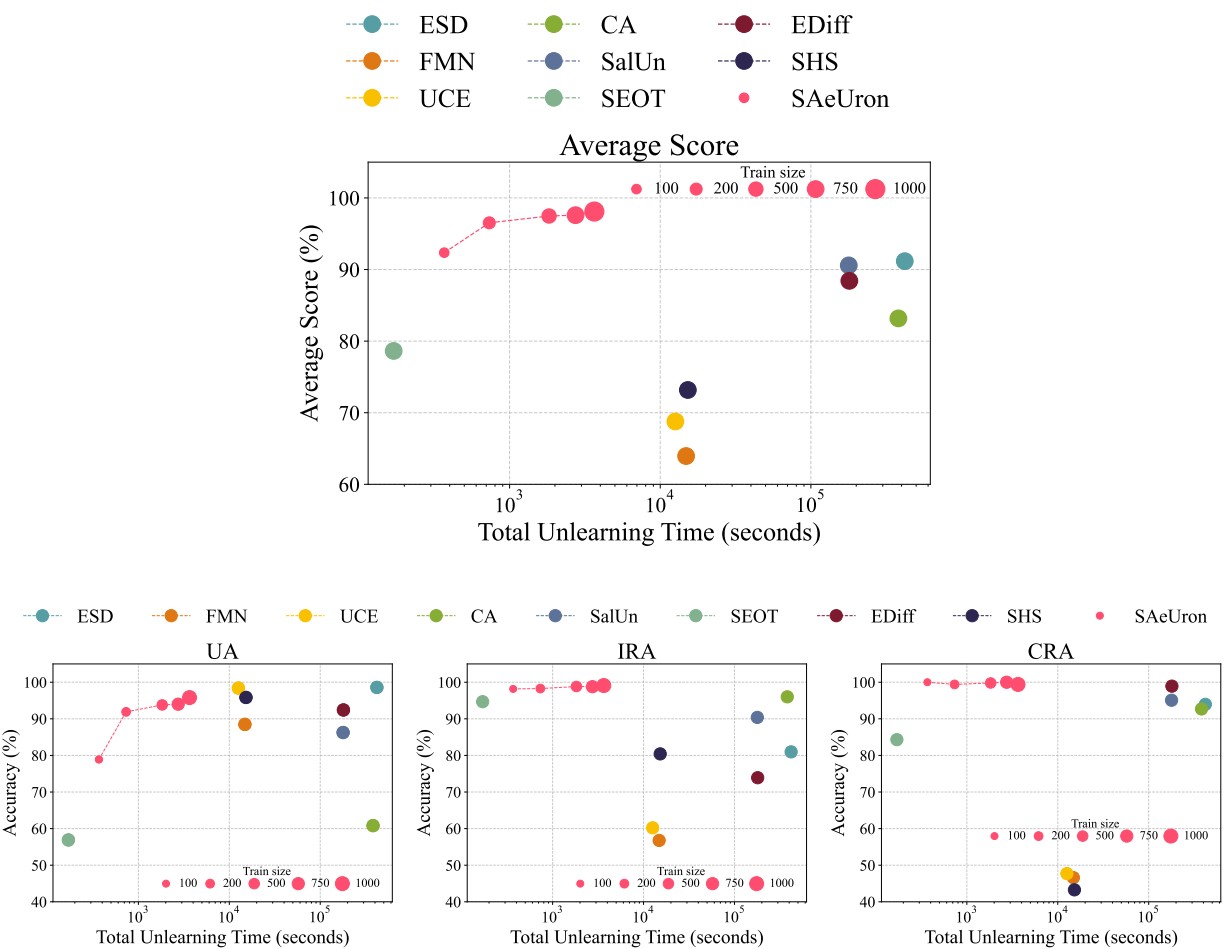

*Figure 23.* **Comparison of total time needed for style unlearning across methods.** We evaluate SAeUron across 5 different sizes of SAE training sets, demonstrating its scalability and high performance even with highly limited data. At the same time, all setups of our approach perform unlearning more efficiently than most methods.

# O. Limitations of SAeUron

The main limitation of our method regarding the unlearning performance can be observed in a situation where two classes - unlearned and remaining ones, share high similarity. In such cases, we might also ablate features that are activated for the remaining class. To visualize this issue, we present generated examples of the dog class while unlearning the cat class and vice versa in Figure 24. Observed degradation is mainly due to the overlap of features selected by our score-based approach during initial denoising timesteps (Figure 25). To further investigate this issue, in Figure 26 we visualize overlapping features. The strength of using SAEs for unlearning is that we can interpret the failure cases of our approach - in this case overlaped feature is related to the generation of heads of both animals. During later timesteps, our score-based selection approach is mainly effective and features that either do not activate at all on the other class or activate with a much smaller magnitude (Figure 27 and Figure 28).

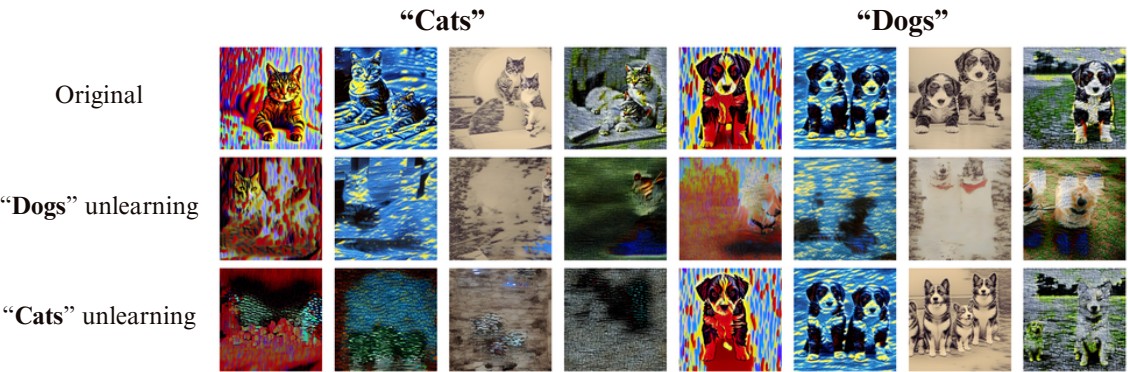

*Figure 24.* **Impact of *Cats* and *Dogs* unlearning on each other.** Due to partially shared features selected for unlearning, we observe a degrading impact of unlearning. *Dogs* unlearning particularly negatively impacts the quality of generated *Cats*.

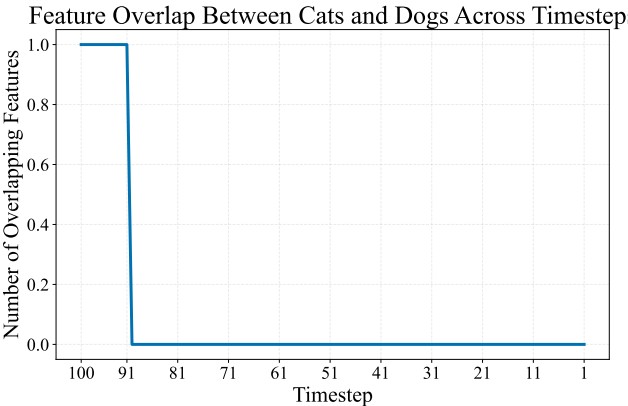

*Figure 25.* **Number of overlapping features selected for unlearning of *Dogs* and *Cats*.** During the first 10 denoising timesteps there is a feature that is selected by our score-based approach for unlearning of both classes. We show activations of this feature in Fig. 26.

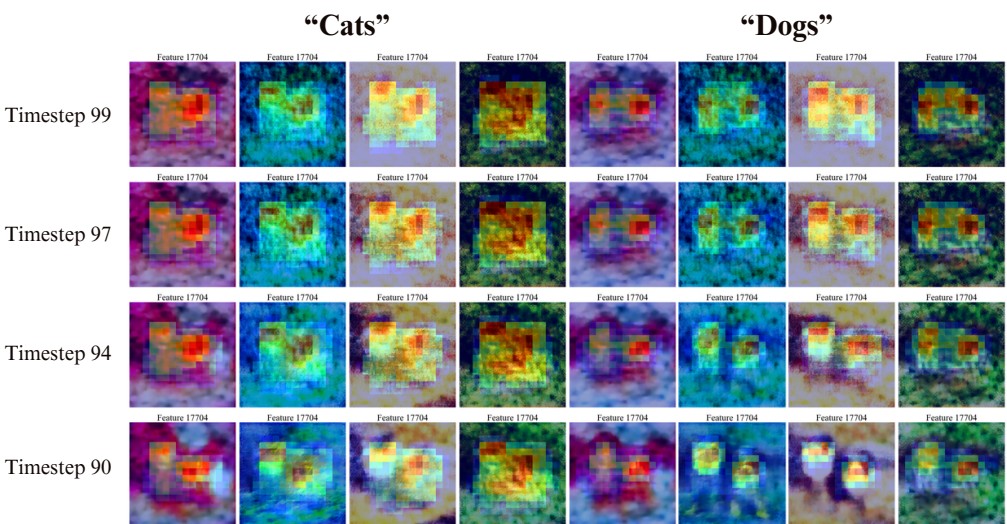

*Figure 26.* **Activations of overlapped feature 17704 during the first denoising steps.** Feature is related to the generation of the head of both animals.

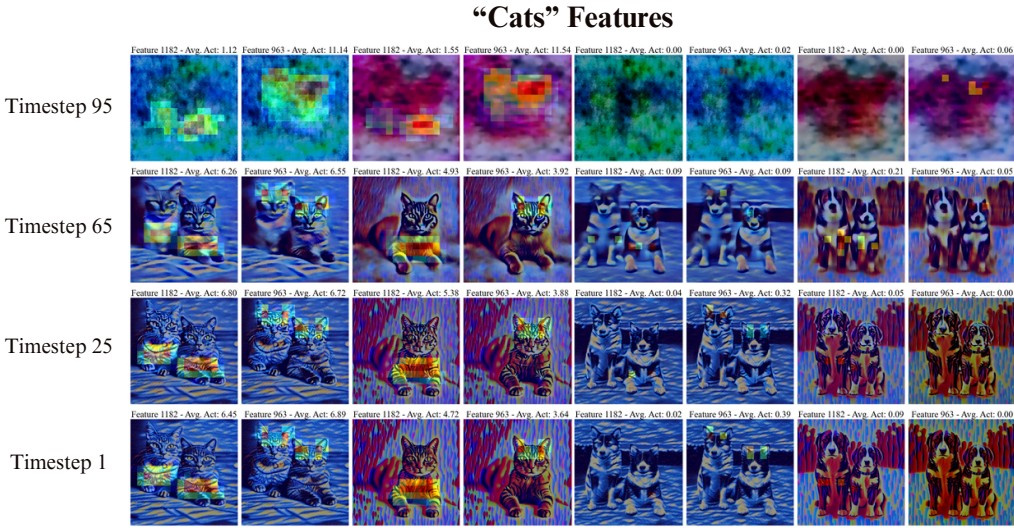

*Figure 27.* **Activations of features selected for unlearning of *Cats* during generation of *Cats* and *Dogs*.** Selected features do not activate on a *Dog* examples at all, or with notably smaller magnitude.

## "Dogs" Features

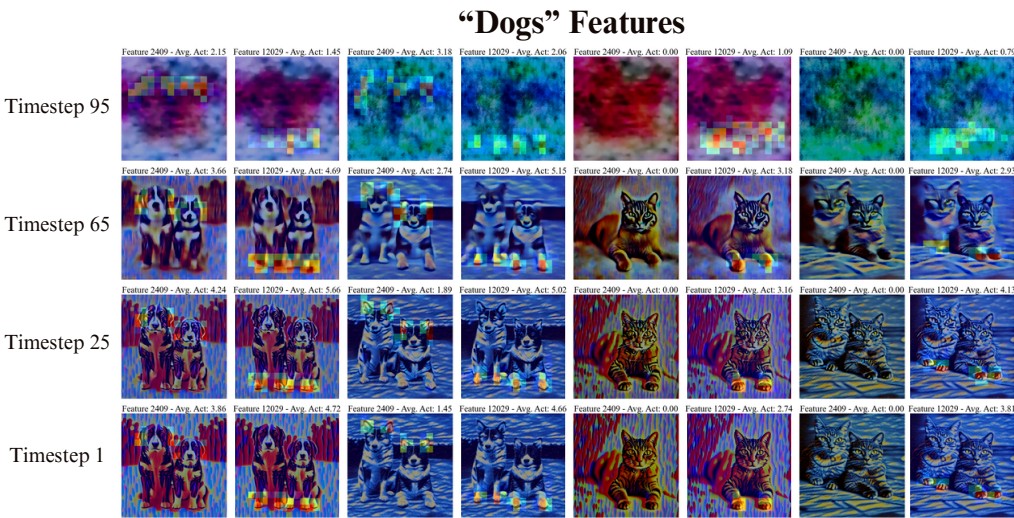

*Figure 28.* **Activations of features selected for unlearning of *Dogs* during generation of *Cats* and *Dogs*.** Selected features do not activate on a *Cat* examples at all, or with notably smaller magnitude.

## O.1. Unlearning of abstract concepts

To assess how SAeUron performs in unlearning broad and more abstract concepts, we evaluated its performance on the full I2P benchmark. Following prior works, we use the Q16 detector to assess whether a generated image contains inappropriate content. The results are presented in Table 8. We observed that, compared to other benchmarks, our method underperforms on this one, performing on par only with the FMN method.

We attribute this outcome to the fact that in SAeUron, we train the SAE on internal activations of the diffusion model. As a result, the learned sparse features correspond to individual visual objects, such as cat ears or whiskers. Thus, while our method effectively removes well-defined concepts composed of visual elements (e.g., nudity or blood), it struggles to capture abstract notions like hate, harassment, or violence.

*Table 8.* **Inappropriate content unlearning evaluation.** Following Schramowski et al. (2023) and Huang et al. (2024), we present the ratio of inappropriate content as a meric. The best result for each metric is highlighted in bold.

| Method | Hate ($\downarrow$) | Harassment ($\downarrow$) | Violence ($\downarrow$) | Self-harm ($\downarrow$) | Shocking ($\downarrow$) | Illegal Activity ($\downarrow$) | Overall ($\downarrow$) | CLIPScore ($\uparrow$) | FID ($\downarrow$) |
|---|---|---|---|---|---|---|---|---|---|
| **FMN** (Zhang et al., 2024a) | 37.7% | 25.0% | 47.8% | 46.8% | 58.1% | 37.0% | 45.4% | 30.39 | 13.52 |
| **SLD-M** (Schramowski et al., 2023) | **22.5%** | 22.1% | 31.8% | 30.0% | 40.5% | 22.1% | 28.2% | 30.90 | 16.34 |
| **ESD-x** (Gandikota et al., 2023) | 26.8% | 24.0% | 35.1% | 33.7% | 40.1% | 26.7% | 31.1% | 30.69 | 14.41 |
| **UCE** (Gandikota et al., 2024) | 36.4% | 29.5% | 34.1% | 30.8% | 41.1% | 29.0% | 33.5% | 30.85 | 14.07 |
| **Receler** (Huang et al., 2024) | 28.6% | **21.7%** | **27.1%** | **24.8%** | **34.8%** | **21.3%** | **26.4%** | 30.49 | 15.32 |
| **SAeUron** | 45.4% | 40.7% | 46.9% | 41.57% | 53.9% | 36.3% | 44.6% | 30.45 | 15.08 |
| **SD** | 44.2% | 37.5% | 46.3% | 47.9% | 59.5% | 40.0% | 46.9% | 31.34 | 14.04 |

# P. Qualitative comparison against other methods

In Figure 29 and Figure 30, we present a qualitative comparison with the best-performing contemporary techniques. In the first set of plots, we show how SAeUron can remove the Bloosom Season style while retaining the original objects and the remaining styles. This is not the case for the other approaches that often fail to generate more complex classes like statues or the sea. At the same time, our technique can unlearn the Dogs class without affecting the remaining objects or all of the evaluated styles.

**Unlearning target: "Blossom Season"**

**Test prompt:** "An image of {object} in Blossom Season style."

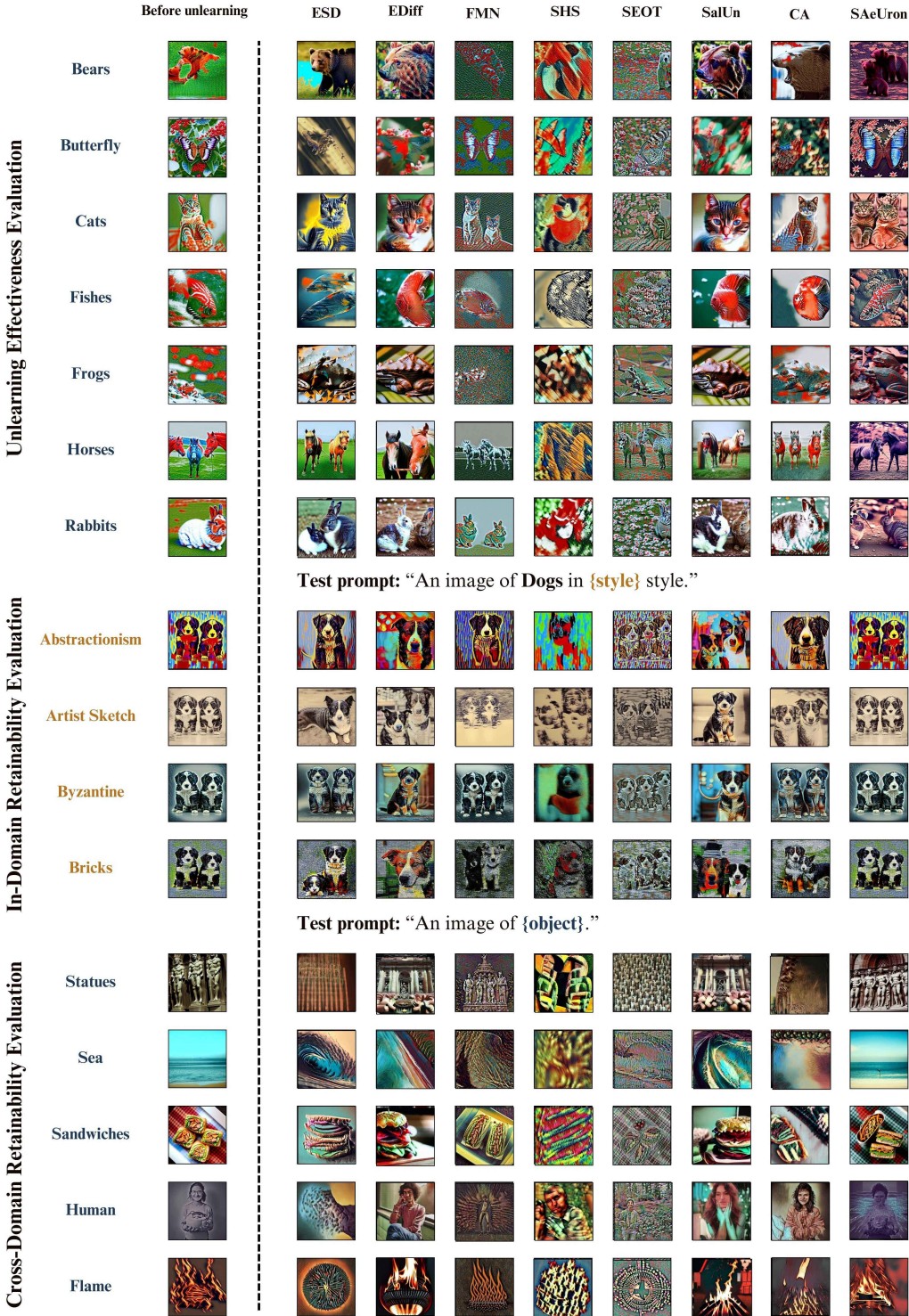

*Figure 29.* **Qualitative comparison with other methods on *style unlearning*.**

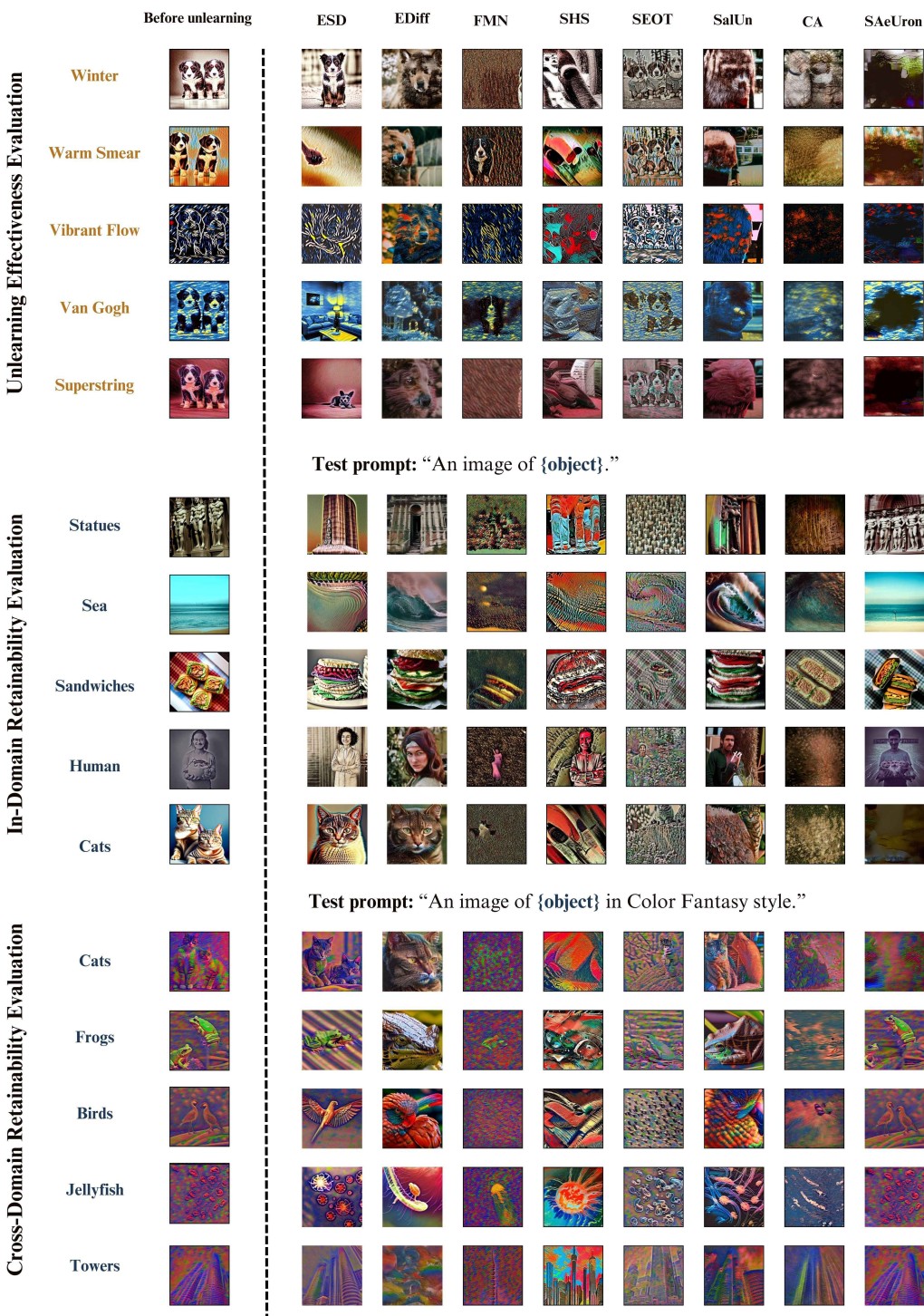

*Figure 30.* **Qualitative comparison with other methods on *object unlearning*.**

# Q. Hyperparameters selection

As stated in Section 5.3, we tune hyperparameters in our validation set. Here, we evaluated multiple values for the multiplier and the number of features selected for unlearning. As shown in Figure Figure 31 of additional results, our method is robust to these parameters, with a broad range of values (apart from extreme cases) yielding comparably high performance.

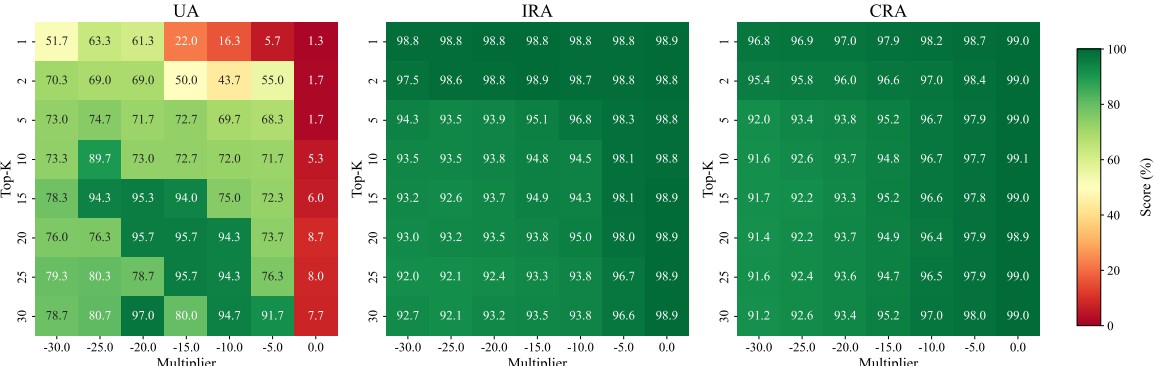

*Figure 31.* **Evaluated metrics for unlearning of classes *Cats*, *Dogs* and *Towers* with different number of selected features and multiplier.**

# R. Detailed performance evaluation

To measure the impact of unlearning on other concepts in Figure 32 we present accuracy on each of 20 classes from UnlearnCanvas benchmark during unlearning. For most classes, our method successfully removes the targeted class while preserving the accuracy of the remaining ones. Nonetheless, in some cases where classes are highly similar to each other (e.g., Dogs and Cats), removing one of them negatively impacts the quality of the other. This observation is consistent across evaluated methods as presented in appendix D.3 of the original UnlearnCanvas benchmark paper.

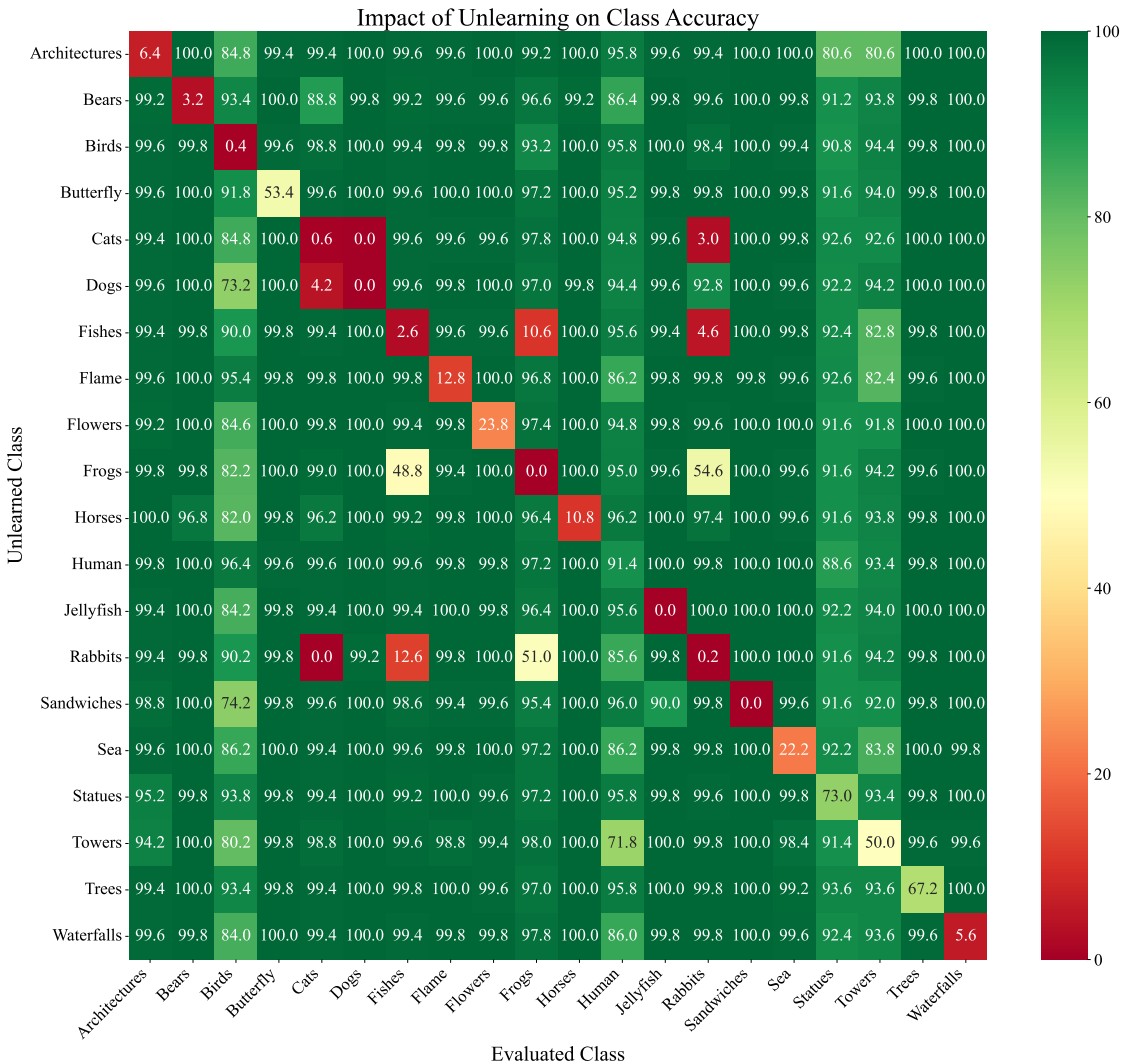

*Figure 32.* **Impact on classification accuracy of classes during unlearning.** Unlearning of similar classes, such as *Cats* and *Dogs*, negatively impacts each other. Complicated classes, such as *Human*, are harder to be effectively removed.

## S. Pseudocode of SAeUron

For ease of understanding our unlearning procedure, we present detailed pseudocode of SAeUron applied on a single denoising timestep $t$ in Algorithm 2.

---

**Algorithm 1** Prepare for unlearning

---

1: **Input:** concept $c$, timestep $t$, dataset $\mathcal{D}$, number of features $\tau_c$, SAE width $n$
2: $\mathcal{F}_c \leftarrow \{i \mid i \in \text{Top-}\tau_c(\{\text{score}(i, t, c, \mathcal{D})\})\}$
3: $\mu(i) \leftarrow \mu(i, t, \mathcal{D}) \quad \forall i \in \mathcal{F}_c$
4: **Output:** $\mathcal{F}_c, \boldsymbol{\mu}$

---

**Algorithm 2** SAeUron unlearning method for DMs

---

1: **Input:** target concept $c$, denoising timestep $t$, validation dataset $\mathcal{D}$, number of features $\tau_c$, multiplier $\gamma_c$
2: $F_t \in \mathbb{R}^{h \times w \times d} \leftarrow$ feature map from cross-attention block
3: $F_t^{\text{flat}} \in \mathbb{R}^{(h \times w) \times d} \leftarrow \text{Flatten}(F_t)$
4: $\hat{F}_t^{\text{flat}} \leftarrow F_t^{\text{flat}}$
5: $\mathcal{F}_c, \boldsymbol{\mu} \leftarrow \texttt{prepare}(c, t, \mathcal{D}, \tau_c)$
6: **for** $j = 1$ **to** $(h \times w)$ **do**
7: $\quad \mathbf{x}^{(j)} \leftarrow F_t^{\text{flat}}[j]$
8: $\quad \mathbf{z}^{(j)}, \hat{\mathbf{z}}^{(j)} \leftarrow \text{TopK}\big(W_{\text{enc}}(\mathbf{x}^{(j)} - \mathbf{b}_{\text{pre}})\big)$
9: $\quad$ **for all** $i \in \mathcal{F}_c$ **do**
10: $\quad\quad$ **if** $\hat{z}_i^{(j)} > \boldsymbol{\mu}_i$ **then**
11: $\quad\quad\quad \hat{z}_i^{(j)} \leftarrow \gamma_c \, \mu\big(i, t, \mathcal{D}_c\big) \, \hat{z}_i^{(j)}$
12: $\quad\quad$ **end if**
13: $\quad$ **end for**
14: $\quad \hat{F}_t^{\text{flat}}[j] \leftarrow W_{\text{dec}} \hat{\mathbf{z}}^{(j)} + \mathbf{b}_{\text{pre}} + \big(\mathbf{z}^{(j)} - \hat{\mathbf{z}}^{(j)}\big)$
15: **end for**
16: $\hat{F}_t \leftarrow \text{Reshape}\big(\hat{F}_t^{\text{flat}}, (h, w, d)\big)$
17: **Output:** $\hat{F}_t \leftarrow$ modified feature map with removed $c$

---

## T. Auto-interpreting features selected for unlearning

To validate whether the features selected by our score-based method correspond to meaningful and interpretable concepts, we construct a simple annotation pipeline using GPT-4o (Hurst et al., 2024). To achieve this, we design a prompt for the GPT model, closely following the one presented in (Paulo et al., 2024) and adapting it to our case. Below, we present this prompt:

```
You are a meticulous AI researcher conducting an important investigation into
visual patterns and feature activations.  Your task is to analyze two sets
of images and provide an explanation that thoroughly encapsulates the visual
features that trigger a particular activation.

You will be presented with two rows of image examples:

Row 1:  Original Images (Context).  This row contains 5 original images.  These
images provide the visual context for the feature analysis.

Row 2:  Activation Overlay Images (Feature Activation).  This row contains 5
images.  Each image in this row corresponds to the image directly above it in
Row 1, but with a visual overlay.  The overlay marks specific regions where a
particular feature is strongly activated in the corresponding original image.

Your goal is to produce a concise, final description that summarizes the shared
visual features and patterns you observe in the highlighted regions of the Row 2
```

(Activation Overlay Images), while using the Row 1 (Original Images) for context. Please adhere to the following guidelines:

Focus on summarizing the visual pattern of activation: Describe the overarching visual features common to the highlighted areas in the Row 2 (Activation Overlay Images). Identify and explain the visual patterns you discern within these overlayed regions. Do not simply describe the entire images in Row 1 or Row 2, but specifically analyze what visual elements within the activated overlays in Row 2, when seen in the context of the corresponding original images in Row 1, indicate the feature is detecting.

Utilize Context from Original Images: Refer to the Row 1 (Original Images) to understand the objects, scenes, or visual elements present in the areas where the feature is activated in Row 2. The original images provide crucial context for interpreting the feature.

Be concise: Keep your final explanation brief and to the point. The explanation should be a single, concise sentence.

Ignore uninformative examples: If some image pairs or their overlays seem unclear or do not contribute to identifying a visual pattern, you may disregard them in your explanation.

Omit marking details: Do not mention the specifics of the visual marking (e.g., "red overlay," "highlighted pixels"). Focus solely on the visual content of the activated regions in Row 2 and describe only the visual content of the activated regions.

Single explanation: Provide only one concise explanation, not a list of possible explanations.

Formatted output: The very last line of your response must be the formatted explanation, beginning with [EXPLANATION]: followed by your concise explanation.

Analyze the following two rows of images (Row 1: Original Images, Row 2: Activation Overlay Images) and provide your formatted explanation:

Alongside the prompt, we provide the GPT-4o model with images from each class in 10 randomly selected styles. The model generates feature annotations separately for each style. Figure 33, Figure 34, and Figure 35 visualize feature activations alongside generated annotations for different objects. As seen in the provided annotations, the GPT model successfully identified the visual features corresponding to the targeted concepts.

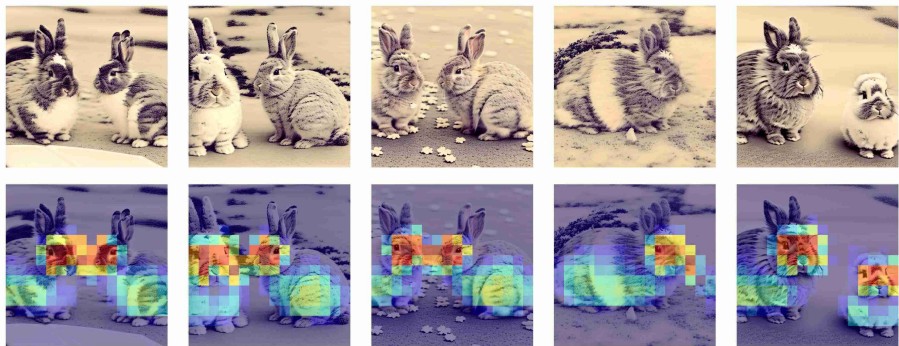

(a) **Generated annotation:** "The activations are strongly triggered by the facial features and eyes of the rabbits."

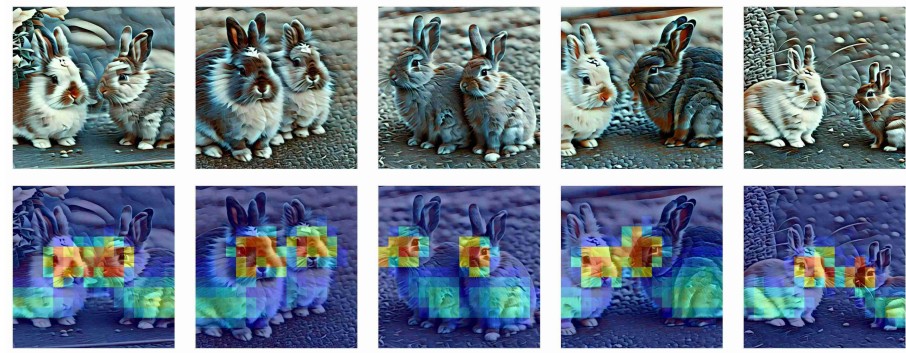

(b) **Generated annotation:** "The feature activates in regions corresponding to the rabbits' facial features and ears, suggesting a focus on these distinct animal characteristics."

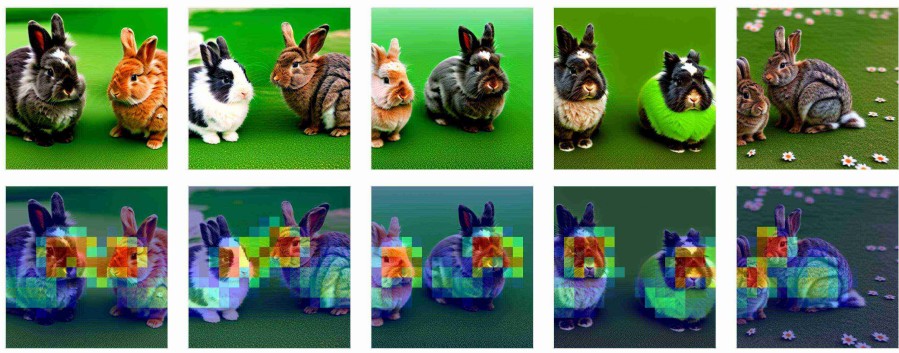

(c) **Generated annotation:** "The activation overlays predominantly highlight the facial features and ear regions of the rabbits, indicating the model is detecting distinct facial and ear characteristics."

*Figure 33.* **Activations of selected features for *Rabbits* unlearning with their annotations generated by the GPT model.**

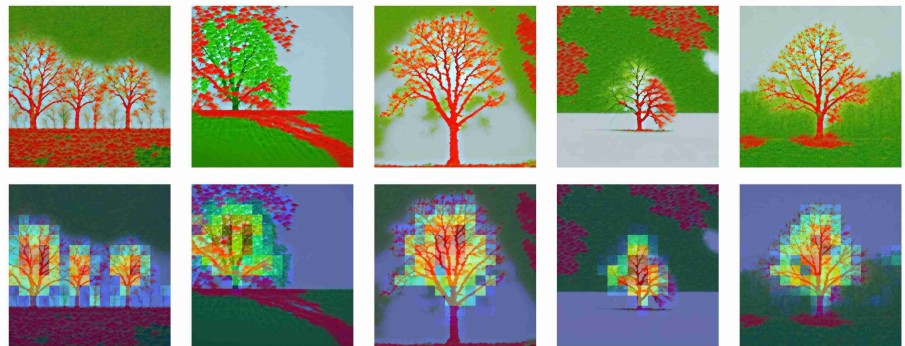

(a) **Generated annotation:** "The activated regions consistently highlight branching structures typical of tree tops, indicating the feature is detecting the intricate patterns of tree branches."

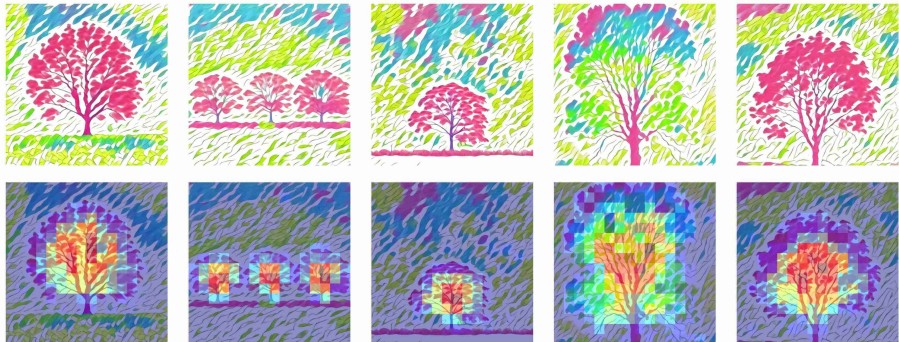

(b) **Generated annotation:** "The feature activation highlights the central shape and structure of the tree tops, focusing on the dense and rounded clusters of foliage."

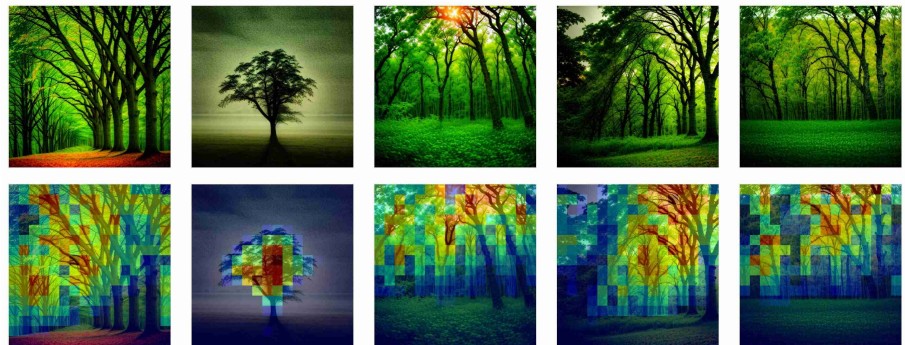

(c) **Generated annotation:** "The activations consistently highlight the vertical and branching structures of trees, capturing the contrast between tree trunks and foliage in various lighting conditions."

*Figure 34.* **Activations of selected features for *Trees* unlearning with their annotations generated by the GPT model.**

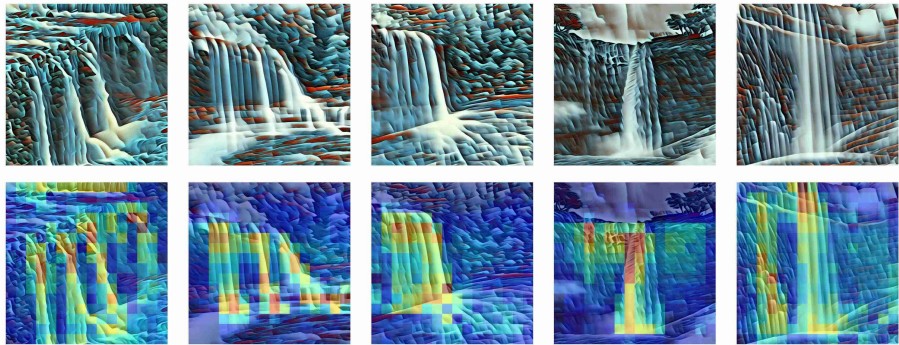

(a) **Generated annotation:** "The feature activation emphasizes vertical flows and streaming patterns resembling waterfalls, highlighting their smooth, elongated shapes and cascading movements against a textured background."

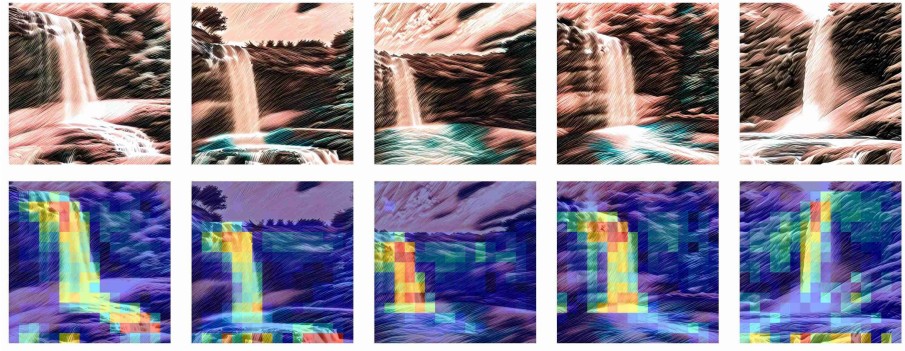

(b) **Generated annotation:** "The feature activation is consistently triggered by the elongated vertical flow and white foamy appearance typical of waterfalls in the images."

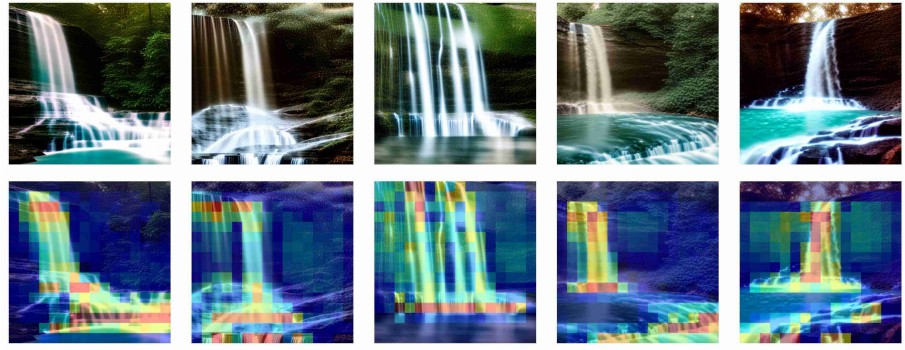

(c) **Generated annotation:** "The visual pattern of activation corresponds to the vertical flow and cascading movement of water in waterfalls, highlighted by the contrast between the bright water streams and the darker, textured background."

*Figure 35.* **Activations of selected features for *Waterfalls* unlearning with their annotations generated by the GPT model.**

