# OpenReview forum: "SAeUron: Interpretable Concept Unlearning in Diffusion Models with Sparse Autoencoders"
_ICML.cc/2025/Conference — ICML 2025 poster_

### Official Review · Reviewer_vmrV · 2025-03-13

**Overall Recommendation:** 4

**Summary:**

This paper introduces SAeUron, a concept unlearning method for text-to-image diffusion models that leverages sparse autoencoder (SAE). The authors first train an SAE on features extracted from the cross‐attention layers and then perform unlearning based on the feature importance scores of specific concepts. During inference, the selected features are modified by applying a negative multiplier while preserving the activations of other concepts. Experimental results demonstrate that the proposed approach outperforms other baselines on style unlearning and maintains robustness after sequential concept removal and under adversarial attacks.

## update after rebuttal
Since most of my questions have been addressed, I have raised my score to 4

**Claims And Evidence:**

The claims are convincing. The authors provide clear evidence showing that the selected concept features are interpretable and that the method achieves state-of-the-art performance compared with existing baselines.

**Essential References Not Discussed:**

N/A

**Experimental Designs Or Analyses:**

**Strength:**

The experimental design is grounded in established benchmarks. The analysis of feature interpretability convincingly demonstrates that the method is able to extract meaningful, concept-specific features from the diffusion model.

**Weakness:**

- It remains unclear whether the removal of one concept (e.g., Husky) may inadvertently affect the generation of similar or neighboring concepts (e.g., Chihuahua). A deeper analysis of collateral damage on related concepts after unlearning would strengthen the study.
- The paper lacks a comprehensive ablation study on the intervention mechanism. Specifically, it is uncertain if the chosen negative multiplier is the optimal way to modify the SAE features or if alternative interventions, such as zeroing out the feature or dynamically set multiplier might yield better results.
-  For practical deployment, it would be valuable to evaluate the method’s performance when a large number of concepts (e.g., all 50 concepts in the UnlearnCanvas benchmark) are removed simultaneously. It is unclear whether the method can retain its performance under such extreme scenarios.

**Methods And Evaluation Criteria:**

The proposed method is well motivated and appears well-suited to the problem of machine unlearning in text-to-image diffusion models. In particular, the use of sparse autoencoders enables the removal of specific concept features while largely preserving other generative capabilities. The evaluation, which employs an existing benchmark for this task, shows promising results that support the effectiveness of the approach.

**Other Comments Or Suggestions:**

N/A

**Other Strengths And Weaknesses:**

The reported 10% computational overhead during inference is relatively high compared with alternative approaches such as ESD or SalUn. This increased overhead may limit the practicality of the method in time-sensitive applications.

**Questions For Authors:**

See Experimental Designs Or Analyses

**Relation To Broader Scientific Literature:**

Although there is extensive literature on mechanistic interpretability using sparse autoencoders, few studies have applied this technique to achieve state-of-the-art results in practical applications. The success of SAeUron in attaining competitive performance is therefore surprising and provides valuable insights into how sparse autoencoders can be utilized in real-world scenarios.

**Theoretical Claims:**

N/A

---

> ### Author Rebuttal · Authors · 2025-03-31
>
> Thank you for the positive review of our work, we would like to explain and address the comments and remaining weaknesses with the help of additional tables and figures provided in the anonymized link [Anonymous github](https://anonymous.4open.science/r/saeuron-8D02/saeuron_rebuttal.pdf):
>
> >The paper lacks a comprehensive ablation study on the intervention mechanism. Specifically, it is uncertain if the chosen negative multiplier is the optimal way to modify the SAE features or if alternative interventions, such as zeroing out the feature or dynamically set multiplier might yield better results.
>
> Thank you for pointing out this omission. Our unlearning method has two main parameters: the number of features selected for unlearning (percentile) and the negative multiplier. We comprehensively analyze their impact in the additional results. As shown in Figure 10, our method is robust to these parameters, with a broad range of values—apart from extreme cases—yielding comparably high performance. Following the reviewer's suggestion, we evaluate the alternative intervention of zeroing out features (when multiplier = 0), observing that we need negative values of the multiplier in order to obtain satisfying unlearning results.
>
> >It remains unclear whether the removal of one concept (e.g., Husky) may inadvertently affect the generation of similar or neighboring concepts (e.g., Chihuahua). A deeper analysis of collateral damage on related concepts after unlearning would strengthen the study.
>
> To provide a more comprehensive analysis of the unlearning effect on similar concepts, in Fig.9, we present how the unlearning of a specific object affects all of the other objects evaluated in the UnlearnCanvas benchmark. We can observe some cases where the unlearning of, for example, *dogs* can lead to the degradation of the quality of *cats*. This observation is consistent across evaluated methods as presented in appendix D.3 of the original [UnlearningCanvas benchmark](https://arxiv.org/abs/2402.11846).
>
> >For practical deployment, it would be valuable to evaluate the method’s performance when a large number of concepts (e.g., all 50 concepts in the UnlearnCanvas benchmark) are removed simultaneously. It is unclear whether the method can retain its performance under such extreme scenarios.
>
> Thank you for this excellent suggestion. We run additional experiments to evaluate such an extreme approach, where we simultaneously unlearn 49 out of 50 styles present in the UnlearnCanvas benchmark (leaving one style out to evaluate the quality of its preservation). For 3 randomly selected combinations, we observe almost no degradation in SAeUron performance. **For unlearning of 49/50 styles simultaneously, we observe UA: 99.29%, IRA: 96.67%, and CRA: 95.00%**. Those results highlight the unprecedented degree of unlearning precision with SAeUron.
>
> >The reported 10% computational overhead during inference is relatively high compared with alternative approaches such as ESD or SalUn. This increased overhead may limit the practicality of the method in time-sensitive applications.
>
> Thank you for highlighting this important limitation of our approach. Your feedback prompted us to re-evaluate our implementation. Since the SAE is a small linear model, its forward pass adds only two matrix multiplications, which should be negligible compared to the diffusion UNet model. Upon further analysis, we discovered that the 10% computational overhead was primarily due to an inefficient implementation: we were redundantly recalculating the same feature selection for unlearning during each pass through the UNet model. By computing this information once and storing it in memory, we have now reduced the computational overhead to just 1.92%.
>
> Thank you for noticing that our method is one of the few studies that have applied SAE to achieve state-of-the-art results in practical applications. To further strengthen this claim, in Tab.2 of additional results, we present how SAeUron can also be used in an even more practical use-case of nudity unlearning, where it also achieves state-of-the-art results.
>
> We hope that our answers provided meaningful clarification to your questions and addressed all of the weaknesses. If true, we kindly ask the reviewer to consider raising their score.

---

> > ### Comment · Reviewer_vmrV · 2025-04-01
> >
> > Thank you for your comments! Most of my questions have been addressed. It is interesting to see that the proposed method is robust to unlearning multiple concepts, the collateral damage is reasonably low, and the actual overhead is quite negligible. I was also surprised that zeroing out the target feature results in poor performance. I am happy to increase my score.

---

### Official Review · Reviewer_8Z9i · 2025-03-14

**Overall Recommendation:** 4

**Summary:**

This paper presents an efficient unlearning framework that leverages sparse auto-encoders to identify relevant features that represent the concepts users want to negate. In previous studies, it was challenging to effectively erase specific concepts while preserving the ability to generate images. This is because unlearning specific concepts leads to modifications in network parameters, resulting in influences on creating images unrelated to that concept. Technically, this paper employs a two-phase approach. In the first phase, the sparse auto-encoder is trained using benchmark datasets developed by [1]. In the second phase, the trained sparse auto-encoder is used to discriminate effective features for unlearning. Empirically, this paper demonstrates that the proposed method achieves comparable results in both concept erasure and maintaining image quality, even with significantly reduced computational resources compared to previous approaches.

[1] Kumari, Nupur, et al. "Ablating concepts in text-to-image diffusion models." CVPR2023.

## **Update after rebuttal**

I appreciate the author’s effort in addressing my remaining concerns. After reviewing their response, which discusses limitations and provides additional experimental results for erasing similar concepts, I am satisfied with the response and the logical approach. Therefore, I decide to increase my score by "Accept".

**Claims And Evidence:**

This paper aims to address the issue of reduced image quality after unlearning specific concepts. While the experimental results show that the numeric values for concept erasure are moderate across all benchmarks, the paper achieves comparable generation quality at the lowest costs, as measured by FID scores. Namely, this paper contributes to the development of the most cost-effective concept erasure method while maintaining image quality.

**Essential References Not Discussed:**

The key contribution of this article lies in its effective and efficient concept erasure technique, which leverages a sparse autoencoder. As a concurrent work, this paper also aims to eliminate unsafe concepts [1]. While it’s understandable that the authors may not need to directly compare their approach with [1], it would be beneficial for them to address different aspects of their methodology.

[1] Kim, Dahye, and Deepti Ghadiyaram. "Concept Steerers: Leveraging K-Sparse Autoencoders for Controllable Generations."arxiv2025.

**Experimental Designs Or Analyses:**

I believe that nudity would be an effective way to showcase its strength.

**Methods And Evaluation Criteria:**

Its experimental design makes sense for concept erasure.

**Other Comments Or Suggestions:**

I believe that the sequence of concepts depends on the difficulty of erasing them. Especially, there might be ambiguous situations where we want to negate “chiwhawha” but preserve “cats” after erasing “tiger.” In this case, I’m curious that sparse autoencoders still manage to raise “chiwhawha” but preserve “cats.” I believe that this paper needs to explore the correlation among multiple concepts and how stable performance is maintained.

**Other Strengths And Weaknesses:**

### **Weakness** ###

- I believe that the two-phase approach is challenging in terms of training time for unlearning. While its efficient algorithm uses minimal computational resources, it becomes impractical if it requires excessive time.
- The authors have presented developing online (streamlined) unlearning approaches that can simultaneously eliminate multiple concepts [1]. I find this capability more appealing than single concept erasure. However, this paper lacks a rigorous explanation for its preservation metrics. Capturing exact features for a specific concept doesn’t necessarily mean direct erasure of that concept stably. Instead, I believe the relative magnitude of the concept compared to others should be considered when choosing an unlearning objective. Nevertheless, this paper emphasizes on numerical performance where the proposed method outperforms baselines in multi-concept erasure.

[1] Lu, Shilin, et al. "Mace: Mass concept erasure in diffusion models."CVPR2024.

**Questions For Authors:**

The paper’s dense content makes it challenging to grasp its main technical message.
I believe that the presentation of numerical superiority doesn’t necessarily imply its worthiness for acceptance at a top-tier conference. In particular, the title suggests that the paper should demonstrate the explainability of unlearning and its relevant features. However, I fail to understand why this paper stands out compared to baselines. While the high-level context appears reasonable, the current presentation lacks a technical explanation for its superiority over baselines. Essentially, this study demonstrates the power of sparse autoencoders and the effectiveness of a linear combination of unlearning.

In the rebuttal, I hope the authors present a comprehensive analysis of the explanability and effectiveness of its approach in ambiguous situations where multiple concepts intersect. If they do so, I will vote for acceptance. However, the current form does not convince to meet the standard of top-tier conferences.

**Relation To Broader Scientific Literature:**

The paper’s key contribution lies in its focus on computation budget, as evident from previous research. This paper presents experimental results that demonstrate the achievement of comparable performance in an efficient manner. The paper’s advantage lies in the use of a sparse auto-encoder to identify effective features releated to specific concepts.

[1] Gandikota, Rohit, et al. "Erasing concepts from diffusion models." CVPR2023.
[2] Gandikota, Rohit, et al. "Unified concept editing in diffusion models." WACV2024.

**Theoretical Claims:**

No discuss

---

> ### Author Rebuttal · Authors · 2025-03-31
>
> Link to additional tables and figures: [Anonymous github](https://anonymous.4open.science/r/saeuron-8D02/saeuron_rebuttal.pdf)
> > **Nudity evaluation**
>
> We thank the Reviewer for suggesting showcasing SAeUron's strength in the real-world use case of unlearning nudity. To do this, we evaluated our method on an established I2P benchmark consisting of 4703 inappropriate prompts.
> We train SAE on SD-v1.4 activations gathered from random 30K captions from COCO train 2014. Additionally, we add to the train set prompts "naked man" and "naked woman" to enable SAE to learn nudity-relevant features. SAE is trained on up.1.1 block, following all hyperparameters used for class unlearning in our submission. We use our score-based method to select features related to nudity, selecting features that strongly activate for "naked woman" or "naked man" prompts and not activating on a subset of COCO train captions.
>
> Following other works, we employ the NudeNet detector for nudity detection, filtering out outputs with confidence less than 0.6. Additionally, we calculate FID and CLIPScore on 30k prompts from the COCO validation set to measure the model's overall quality when applying SAE. As shown in Tab 2, SAeUron achieves state-of-the-art performance in removing nudity while preserving the model's overall quality. This highlights the potential of our method in real-world applications.
>
> > Relation to [1]
>
> Thank you for pointing out this interesting paper. We respectfully note that **it has been submitted to arxiv after the conference deadline**. Nonetheless, while both works intervene on SAE's latent space to remove undesirable concepts, there is a crucial difference in that [1] trains SAE on activations of the text encoder while we directly apply our approach to the diffusion UNet. Since both works utilize SAEs for similar use cases, we will gladly add this paper as concurrent work in the final version of our submission.
>
> [1] Kim, Dahye, and Deepti Ghadiyaram. "Concept Steerers: Leveraging K-Sparse Autoencoders for Controllable Generations."arxiv2025.
>
> > I believe that the two-phase approach is challenging in terms of training time for unlearning.
>
> To showcase the efficiency of our approach, we measure the time needed to present the results of unlearning in Figs 11 and 12. We evaluate the scaling of the SAeUron approach using the training set of sizes: 100, 200, 500, 750, and 1000 images. We train our SAE for 5 epochs for each scenario, keeping the hyperparameters constant. Our approach achieves good unlearning results even in limited training data scenarios while being more efficient from all methods requiring fine-tuning.
>
> > Online (streamlined) unlearning is appealing but lacks a rigorous explanation for the preservation metrics.
>
> In our main experiments, we stricktly follow the evaluation provided by the UnlearnCanvas benchmark where independently trained classifiers are provided to evaluate *In-domain retain accuracy (IRA)* that measures the quality of generation of other concepts when unlearning a particular one (e.g., 49 other styles when unlearning a single one), and *cross-domain retain accuracy (CRA)* which assesses the preservation quality in a different domain (e.g., in style unlearning, we calculate object classification accuracy). Additionaly, following instructions from the benchmark, we report the FID metric when generating all of the concepts except for the unlearned ones.
>
> > I believe that this paper needs to explore the correlation among multiple concepts and how stable performance is maintained.
>
> We agree with the Reviewer that capturing concept-specific features doesn't directly imply precise unlearning. To measure the impact of unlearning on other concepts in Fig 9 we present accuracy on each of 20 classes from UnlearnCanvas benchmark during unlearning. For most classes, our method successfully removes the targeted class while preserving the accuracy of the remaining ones. Nonetheless, in some cases where classes are highly similar to each other (e.g., Dogs and Cats), removing one of them negatively impacts the quality of the other. This observation is consistent across evaluated methods as presented in appendix D.3 of the original [UnlearningCanvas benchmark](https://arxiv.org/abs/2402.11846).
>
> To study how well the stable performance of our method can be maintained, we run an additional experiment where we subsequently unlearn 49 out of 50 styles present in the UnlearnCanvas benchmark (leaving one style out to evaluate the quality of its preservation). We observe almost no degradation in SAeUron performance for three randomly selected combinations. **For unlearning of 49/50 styles simultaneously, we observe UA: 99.29%, IRA: 96.67%, and CRA: 95.00%**. Those results highlight the unprecedented degree of unlearning precision with SAeUron.
>
> We hope our responses have clarified your questions and addressed any concerns. If so, we would appreciate your consideration in raising the score.

---

> > ### Comment · Reviewer_8Z9i · 2025-04-01
> >
> > I appreciate the author addressing my primary concerns in their rebuttal. However, I have a question about a scenario where multiple concepts are being erased.
> >
> > As you mentioned [UnlearningCanvas Benchmark](https://arxiv.org/pdf/2402.11846), I would like to understand which situations the proposed method negatively impacts. However, all the cases the authors presented are favorable.
> >
> > Since this paper employs a two-phase training approach, it should clarify its strengths and limitations in multiple concept erasure in a row, especially in light of the progress in training-free approaches that focus on one concept erasure at a time.
> >
> > I also have a minor question. Could you provide a explanation of the major differences between Pre-ASR and Post-ASR?
> > I appreciate the author’s rebuttal, as it addresses most of my concerns. I would like to see its limitations and failure cases in addition to the reasons and analysis behind this phenomenon. Once I observe that, I will be happy to increase the score.
> >
> > ———— Post authors response ————
> >
> >
> > I appreciate the author’s effort in addressing my remaining concerns. After reviewing their response, which discusses limitations and provides additional experimental results for erasing similar concepts, I am satisfied with the response and the logical approach. Therefore, I decide to increase my score by "Accept".

---

> > > ### Author Response · Authors · 2025-04-03
> > >
> > > We are happy to know that we have addressed your primary concerns with our rebutal. Here we address the remaining questions:
> > >
> > > > As you mentioned UnlearningCanvas Benchmark, I would like to understand which situations the proposed method negatively impacts. However, all the cases the authors presented are favorable.
> > >
> > > As we presented in Figure 9 of the additional results [Anonymous github](https://anonymous.4open.science/r/saeuron-8D02/saeuron_rebuttal.pdf), the main limitation of our method regarding the unlearning performance can be observed in a situation where two classes - unlearned and remaining ones, share high similarity. In such cases, **we might also ablate features that are activated for the remaining class**. To visualize this issue, we present generated examples of the dog class while unlearning the cat class and vice versa in Fig 12. Observed degradation is mainly due to the overlap of features selected by our score-based approach during initial denoising timesteps (Fig 13).
> > > To further investigate this issue, in Fig 14 we visualize overlapping features. The strength of using SAEs for unlearning is that we can interpret the failure cases of our approach - in this case overlaped feature is related to the generation of heads of both animals.
> > > During later timesteps, our score-based selection approach is mainly effective and features that either do not activate at all on the other class or activate with a much smaller magnitude (Fig 15 and 16).
> > >
> > > > Since this paper employs a two-phase training approach, it should clarify its strengths and limitations in multiple concept erasure in a row, especially in light of the progress in training-free approaches that focus on one concept erasure at a time.
> > >
> > > The two-phase approach that we employ in our work brings both strengths and limitations. Most importantly, in order to maintain high-quality retention of the remaining concepts, we have to train SAE on activations gathered from a reasonable number of various data samples (see Figure 11 in additional results for more details). This might bring some computational overhead when trying to unlearn a single concept, especially when compared with training-free approaches. On the other hand, such an approach naturally allows us to efficiently unlearn several concepts at the same time without the need for any additional training. This also includes concepts not present in the SAE training set, as validated in Appendix F of our original submission.
> > > Another limitation when compared to finetuning-based unlearning is that our solution can only be employed in practice in a situation where users do not have direct access to the model, as it would be relatively easy to remove the blocking mechanism in the open-source situation.
> > > We will add this description to the Limitation section of our revised paper.
> > >
> > > > Could you provide a explanation of the major differences between Pre-ASR and Post-ASR?
> > >
> > > Pre-ASR and Post-ASR are Attack Success Rates of nudity generation directly taken from the official [UnlearnDiffAtk Benchmark](https://huggingface.co/spaces/Intel/UnlearnDiffAtk-Benchmark). Both metrics are measured based on the same set of predefined 143 prompts generating nudity in the base SD-v1.4 model.
> > > Pre-ASR measures the percentage of nudity generated by the unlearned evaluated model. In the Post-ASR scenario, each prompt is additionally tuned in an adversarial way by the UnlearnDiffAtk method to enforce the generation of nudity content. Substantial differences between those two metrics for some methods witness the fact that they are highly vulnerable to this type of attack. Notably, our method achieves a Post-ASR of 1.4\%, yielding the smallest difference between Pre and Post ASR.
> > >
> > > We hope that with the answers mentioned above, we managed to clearly describe the limitations of our work. Please note, that due to ICML limitations, we will not be able to respond to any further questions, but we are thankful to the reviewer for insightful suggestions that enabled us to improve our work.

---

### Official Review · Reviewer_gT3h · 2025-03-14

**Overall Recommendation:** 3

**Summary:**

This paper introduces SAeUron, a novel method for concept unlearning in diffusion models by manipulating intermediate features using Sparse Autoencoders. The Sparse Autoencoder is trained to learn representations where most features have near-zero values, allowing specific concept-related features to be identified. The autoencoder's input consists of intermediate UNet features extracted at certain timesteps during diffusion.

After training is complete, the activated features corresponding to a specific concept (e.g., a class) are identified. During inference, the unlearning process is applied by negating only these specific feature values. The modified features are then passed through the decoder of the Sparse Autoencoder, altering the features of the U-Net to effectively suppress the generation of the targeted concept.

On the UnlearnCanvas benchmark, SAeUron outperforms existing unlearning baselines in style unlearning and achieves comparable results in object unlearning. Additionally, it offers two key advantages over other methods: i) Sequential Unlearning – It performs well when unlearning multiple concepts in sequence. ii) Robustness to Adversarial Prompts – It effectively resists adversarial prompts crafted using the UnlearnDiffAtk method.

## Final recommendation and Justification (post rebuttal)

Thank you for addressing my concerns and answering the questions. The underperformance of SAeUron on broader concepts (e.g., hate) and the degradation of dogs when unlearning cats make sense, and I agree that it's valuable to highlight this in the discussion or limitations section.

The additional experiments presented in the rebuttal have strengthened the paper with a more thorough analysis. I expect the authors to include all of these insights in the camera-ready version. I’m happy to raise my score to Weak Accept.

**Claims And Evidence:**

The authors state, "Evaluation with the competitive UnlearnCanvas benchmark on object and style unlearning highlights SAeUron’s state-of-the-art performance." However, Table 1 shows that SAeUron only achieves the best score in In-domain Retain Accuracy (IRA) and average performance of style unlearning. This does not fully support the claim that SAeUron achieves state-of-the-art results in object and style unlearning.

The authors claim that SAeUron is robust to adversarial attacks, arguing that other methods provide only a limited understanding of base model changes and fail to fully remove targeted concepts. However, Table 1 indicates that SalUn achieves better object unlearning and FID scores than SAeUron, raising questions about its comparative effectiveness. Additionally, the paper lacks qualitative comparisons with baselines to demonstrate that SAeUron selectively removes only the targeted concepts, unlike base models.

While Figure 8 suggests strong resistance to adversarial attacks such as UnlearnDiffAtk, Figure 18 does not show similar robustness for object unlearning attacks. The authors attribute this discrepancy to the evaluation method used for object unlearning but do not clarify whether object unlearning results were excluded in Figure 8, leading to uncertainty about the experimental setup. Furthermore, Table 5 in the appendix presents CLIPScore results before and after a successful adversarial attack but lacks baseline comparisons, making it difficult to determine whether SAeUron outperforms other methods, especially in adversarial attacks.

**Essential References Not Discussed:**

The paper tackles the robustness from the adversarial attacks, yet they did not include the baselines rather recent and specifically robust to those methods [B,C,D]

[B] Shilin Lu, Zilan Wang, Leyang Li, Yanzhu Liu, and Adams Wai-Kin Kong. Mace: Mass concept erasure in diffusion models. CVPR, 2024.

[C] Chi-Pin Huang, Kai-Po Chang, Chung-Ting Tsai, Yung-Hsuan Lai, and Yu-Chiang Frank Wang. 2023. Receler: Reliable concept erasing of text-to-image diffusion models via lightweight erasers. ECCV 2024.

[D] Chao Gong, Kai Chen, Zhipeng Wei, Jingjing Chen, Yu-Gang Jiang. Reliable and Efficient Concept Erasure of Text-to-Image Diffusion Models, ECCV 2024

**Experimental Designs Or Analyses:**

1. **Choice of Blocks**: The decision of where to apply the Sparse Autoencoder (SAE) is critical. The authors mention that they empirically modified the cross-attention layers in the Up block, but the rationale behind prioritizing this block over others remains unclear. While the reviewer acknowledges that the choice of cross-attention blocks is based on prior work (Basu et al., 2024), the authors also highlight distinct attribute-specific differences in causal state distribution (e.g., style being more relevant in self-attention blocks). Additionally, previous research [A] suggests that self-attention layers in the Up block are particularly effective for preserving style. This raises an important question: Would modifying different blocks (e.g., Up, Mid, Down) or targeting self-attention layers instead of cross-attention layers result in better performance? Performing an ablation study on this aspect, beyond just qualitative ablation results, would further strengthen the argument.
2. **Lack of Qualitative Examples**: Since this is a generation-focused paper, it is essential to provide substantial qualitative examples covering a wide range of results. However, the number of provided visual examples is limited, and there are no qualitative comparisons with other baselines. Without such comparisons, it is difficult to assess how SAeUron performs relative to existing methods in terms of preserving image quality while achieving unlearning.
3. **Choice of Thresholds**: While SAeUron enables controlled feature modification through percentile-based selection, there is no ablation study examining its effectiveness in determining the optimal level of change. Specifically, ablations on different threshold values are missing, and the selection is stated to be empirical rather than systematically justified. Without such an analysis, it remains unclear how well SAeUron balances concept removal and image preservation, making its robustness to adversarial attacks less substantiated.

**References**
[A] Jaeseok Jeong, Junho Kim, Yunjey Choi, Gayoung Lee, Youngjung Uh, Visual Style Prompting with Swapping Self-Attention, arxiv 2024

**Methods And Evaluation Criteria:**

The proposed method is primarily evaluated on object and style unlearning. However, unlearning is particularly crucial for NSFW content, which aligns more with concept unlearning rather than just object or style removal. It is also essential for protecting portrait rights. Since the authors do not assess their method on NSFW content, sensitive concepts, or portrait rights, it remains unclear how effectively SAeUron addresses real-world unlearning challenges. This gap in evaluation limits the method's applicability to practical scenarios where unlearning is most critical.

**Other Comments Or Suggestions:**

no

**Other Strengths And Weaknesses:**

Compared to other existing unlearning approaches, this paper addresses the unlearning problem using an autoencoder that operates independently of diffusion models, which is a notable strength.

My major concern is the lack of exploration of experimental settings and insufficient experimental results.  Please see Claims And Evidence* and Experimental Designs Or Analyses*.

The authors did not specifically mention which Stable Diffusion model version was used in this paper.

**Questions For Authors:**

**Practical Applications**: The paper primarily focuses on object and style unlearning, but practical applications of unlearning extend to sensitive content removal, such as NSFW content or specific individuals in generated images. Given the increasing importance of content moderation in generative models, an evaluation on concept unlearning for NSFW or identity removal would better demonstrate the real-world applicability of SAeUron. Have the authors considered these use cases, and how well does the method perform in such scenarios?


**Efficiency**: The authors think it is an efficient method? Methods like SLD or Receler do not require additional training or require only lightweight training, whereas SAeUron involves learning whole autoencoder networks, which may introduce overhead. Additionally, since the threshold selection is empirical for specific object (Table 4), how does this impact efficiency?


**Generalization to Out-of-Distribution (OOD) Data**: Table 3 suggests that SAeUron generalizes well to OOD settings, but the underlying reason for this generalization is not well explained. What specific properties of SAeUron contribute to its robustness in OOD scenarios? Furthermore, how do other benchmark methods perform under the same experimental setup?

**Relation To Broader Scientific Literature:**

The proposed method builds on key existing research, particularly Sparse Autoencoders and Diffusion Models. A key contribution is demonstrating that unlearning in diffusion models can be achieved using an Autoencoder without requiring full fine-tuning of the entire model. This approach is both simple and intuitive, making it explainable compared to traditional fine-tuning methods. This level of interpretability makes it particularly valuable for controlling.

**Theoretical Claims:**

There are no theoretical claims.

---

> ### Author Rebuttal · Authors · 2025-03-31
>
> Link to additional tables and figures: [Anonymous github](https://anonymous.4open.science/r/saeuron-8D02/saeuron_rebuttal.pdf)
> > **Unlearning of nudity**
>
> To highlight the potential of SAeUron in real-world applications, we extend our study to the evaluation with the I2P benchmark focusing on the real-world application of NSFW content removal (see reply to rev 8z9i for technical details). As presented in Table 2, SAeUron achieves the best score in nudity content removal while maintaining high-quality generations for the remaining concepts.
>
> >**Lack of qualitative comparisons with baselines**
>
> In Fig 1 and Fig 2, we present a qualitative comparison with the best-performing contemporary techniques. In the first set of plots, we show how SAeUron can remove the *Cartoon* style while retaining the original objects and the remaining styles. This is not the case for the other approaches that often fail to generate more complex classes like statues or the sea. At the same time, our technique can unlearn the bear class without affecting the remaining objects or all of the evaluated styles.
>
> >SAeUron only achieves the best score in (IRA) and average performance of style unlearning. This does not fully support the claim that SAeUron achieves state-of-the-art results.
>
> Thank you for pointing out the inaccuracy; we will rewrite the sentence in the final version of our paper to highlight that our approach yields state-of-the-art results for style unlearning while scoring the second-best score for objects. The evaluation metrics are designed in a contradictory way, so it is easy to improve on one metric at the expense of another. Therefore, we report the average performance across all three metrics to enable a fairer comparison among all evaluated methods.
>
> >**Performance against adversarial attacks**
>
> As requested, we run additional comparisons with adversarial attacks using the 143 nudity prompts prvided by the authors of UnlearnDiffAtk as a benchmark. As presented in Tab 3, **SAeUron outperforms even the methods highlighted by the reviewer as specifically designed for robustness against adversarial attacks.**
>
> >Selection of the targetted block
>
> The selection of the appropriate activations for applying SAE is a critical decision. To simplify this process, we introduce a straightforward methodology in which we ablate the diffusion blocks one by one to identify the one essential for generating the desired concepts, as detailed in Appendix C. Although this heuristic may not always pinpoint the optimal location for the SAE application, it allowed us to achieve high-quality results.
>
> > Choice of hyperparameters
>
> As stated in Section 5.3.2, we tune hyperparameters in our validation set. Following the reviewer’s suggestion, we evaluated multiple values for the multiplier $\gamma$ and the number of features selected for unlearning (instead of percentile $\tau$, to make the analysis clearer). As shown in Figure 10 of additional results, our method is robust to these parameters, with a broad range of values (apart from extreme cases) yielding comparably high performance.
>
>
> > **Method Efficiency** Methods like SLD or Receler do not require additional training or require only lightweight training, whereas SAeUron involves learning whole autoencoder networks, which may introduce overhead?
>
> We would like to point out that sparse autoencoders used by our method are simple linear models consisting of two matrices. Therefore, their training can be considered lightweight. To prove those claims, in Fig 11 and 12, we compare the unlearning time for different approaches with their scores. We evaluate SAeUron with different training set sizes, showing that our method is faster than almost all evaluated approaches while yielding higher performance as measured by the averaged score.
>
> > **Generalization to OOD**: What specific properties of SAeUron contribute to its robustness in OOD scenarios? Furthermore, how do other benchmark methods perform under the same experimental setup?
>
> In this scenario, we train Sparse Autoencoder on the limited set of styles available in the UnlearnCanvas dataset. Then, we unlearn OOD styles not seen by the SAE during training. This could be understood as a variant of *zero-shot unlearning* where we select features to block only from those already available in SAE without any additional training. Since SAEs are trained in an unsupervised way, they learn an overcomplete set of features, activating also on samples not presented to the model during training. This is in contrast to other methods, where targets need to be explicitly provided during finetuning. As presented in Tab. 3, our method can still achieve 51% unlearning accuracy, with a limited drop in retention accuracy. To our knowledge, none of the evaluated methods could be directly used in such a scenario.
>
> We hope our responses have clarified your questions and addressed any concerns. If so, we would appreciate your consideration in raising the score.

---

> > ### Comment · Reviewer_gT3h · 2025-04-04
> >
> > I appreciate that the authors have addressed most of my concerns. However, I have a few follow-up questions and remaining concerns:
> > 1. Regarding NSFW filtering in the I2P dataset, I wonder whether broader concepts such as hate, harassment, violence, etc can also be effectively erased. I wonder whether a single threshold is sufficient for handling such broad concepts.
> > 2. The qualitative comparison results remain insufficient. In the rebuttal, the authors only present a single object (e.g., bear) and a single style (e.g., cartoon) example to demonstrate unlearning effectiveness. In contrast, most existing works—such as MACE and Receler—provide qualitative comparisons across at least five concepts. SAeUron primarily focuses on showcasing its own successful qualitative results, rather than providing direct comparisons with other methods, which does not clearly convey its relative effectiveness.
> > 3. As Reviewer 8Z9i noted, I am also curious about scenarios in which the proposed method underperforms or has negative impacts compared to other baseline models.
> > 4. The authors did not specifically mention which Stable Diffusion model version was used in this paper, and it would be good to add that.
> >
> > I will revisit my score after reading the authors’ responses. I look forward to their thoughts and would be happy to consider increasing my score if these concerns are adequately addressed.

---

> > > ### Author Response · Authors · 2025-04-07
> > >
> > > We are happy to read that our response addressed most of the reviewers' concerns. Below, we clarify the remaining ones:
> > >
> > > > Regarding NSFW filtering in the I2P dataset, I wonder whether broader concepts such as hate, harassment, violence, etc can also be effectively erased. I wonder whether a single threshold is sufficient for handling such broad concepts.
> > >
> > > Thank you for this excellent suggestion for an additional evaluation. To assess how SAeUron performs in unlearning such broad concepts, we evaluated its performance on the full I2P benchmark. Following prior works, we use the Q16 detector to assess whether a generated image contains inappropriate content. The results are presented in Table 5 of the additional results. We observed that, compared to other benchmarks, our method underperforms on this one, performing on par with the FMN method.
> > >
> > > We attribute this outcome to the fact that in SAeUron, we train the SAE on internal activations of the diffusion model. As a result, the learned sparse features correspond to individual visual objects, such as cat ears or whiskers (see Figures 3 to 6 in the additional results). Thus, while our method effectively removes well-defined concepts composed of visual elements (e.g., *nudity* or *blood*), it struggles to capture abstract notions like *hate*, *harassment*, or *violence*. We will include this evaluation in the camera-ready version of our submission, along with a discussion in the limitations section.
> > >
> > > > The qualitative comparison results remain insufficient. In the rebuttal, the authors only present a single object (e.g., bear) and a single style (e.g., cartoon) example to demonstrate unlearning effectiveness. In contrast, most existing works—such as MACE and Receler—provide qualitative comparisons across at least five concepts. SAeUron primarily focuses on showcasing its own successful qualitative results, rather than providing direct comparisons with other methods, which does not clearly convey its relative effectiveness.
> > >
> > > > As Reviewer 8Z9i noted, I am also curious about scenarios in which the proposed method underperforms or has negative impacts compared to other baseline models.
> > >
> > > In addition to the previously presented examples, we extended the qualitative comparison to more challenging styles (Blossom Season – Fig. 17) and highly interfering objects (Dogs – Fig. 18). We emphasize that our extensive qualitative evaluation—unlike in works such as Receler—includes all aspects of unlearning evaluation for *all methods*, covering both unlearning accuracy and retainability.
> > >
> > > Notably, unlearning the *Dogs* class leads to significant degradation in similar classes such as *Cats*, as shown in Fig. 18 and further analyzed in Fig. 9. Thanks to the interpretability of the independent SAE features used for unlearning, we can pinpoint the reasons for poor performance in such cases.
> > >
> > > To illustrate this issue, we present generated examples of the *Dog* class when unlearning *Cats* and vice versa in Fig. 12. The observed degradation stems from the overlap of features selected by our score-based approach during early denoising steps (Fig. 13). To investigate further, we visualize the overlapping features in Fig. 14. A key strength of using SAEs for unlearning is that we can interpret such failure cases—in this instance, the overlapping feature relates to generating the heads of both animals. In later timesteps, our score-based selection is more effective, isolating features that either do not activate at all or activate only weakly on the other class (Figs. 15 and 16).
> > >
> > > >The authors did not specifically mention which Stable Diffusion model version was used in this paper, and it would be good to add that.
> > >
> > > As stated in Sec. 5.1, for the main experiments we use the UnlearnCanvas benchmark, which provides a fine-tuned version of Stable Diffusion v1.5, available in the [official repository](https://github.com/OPTML-Group/UnlearnCanvas). For NSFW experiments, we follow related works such as MACE and Receler and use SD v1.4.
> > >
> > > >I will revisit my score after reading the authors’ responses. I look forward to their thoughts and would be happy to consider increasing my score if these concerns are adequately addressed.
> > >
> > > We thank the reviewer for the detailed review and comments, which greatly helped us to further improve our work. We believe that our answers above address the reviewer's questions and concerns, and we would greatly appreciate it if you would consider raising the score. As a reminder, due to ICML limitations, we will not be able to respond to any additional comments or questions.

---

### Official Review · Reviewer_WhaF · 2025-03-17

**Overall Recommendation:** 4

**Summary:**

The paper proposed a method of unlearning, i.e., erasing concepts as conditional prompts, in diffusion models. The idea is to represent the concept features in a sparse auto-encoder to compress them into low dimension, then modifies the weights of concept-related features after detecting them, leading to modified generative outputs.

**Claims And Evidence:**

The authors carried out experiments to demonstrate the efficiency of erasing concept-based objects from images, and compare them with previous erasing (unlearning) methods.

**Essential References Not Discussed:**

N/A

**Experimental Designs Or Analyses:**

Yes.

**Methods And Evaluation Criteria:**

The evaluation criteria and benchmarks makes sense.

**Other Comments Or Suggestions:**

N/A

**Other Strengths And Weaknesses:**

The strong point is that the paper have compared their method with a significant number of recent methods. The weak point is that the distinctive idea between the proposed method versus existing ones is not clear stated.

**Questions For Authors:**

When talking about unlearning, a key issue is whether the erasing affects other output, especially similar concepts or objects. The paper has mentioned retention accuracy in the appendix but did not clearly state the evaluation criteria and the meaning of the results. Please elaborate it.

**Relation To Broader Scientific Literature:**

N/A

**Theoretical Claims:**

No theoretical analysis provided.

---

> ### Author Rebuttal · Authors · 2025-03-31
>
> Thank you for the positive review of our work, below we would like to explain and address the comments and remaining weaknesses with the help of additional tables and figures provided in the anonymized link [Anonymous github](https://anonymous.4open.science/r/saeuron-8D02/saeuron_rebuttal.pdf):
>
>
> >The weak point is that the distinctive idea between the proposed method versus existing ones is not clear stated.
>
> Thank you for pointing out this limitation; we will emphasize the distinctiveness of our method when compared with existing solutions in the final version of our submission. Below, we briefly outline the key differences.
>
> Most existing approaches to machine unlearning in diffusion models (e.g., EDiff, ESD, FMN, SalUn, SHS, SA, CA) follow a common principle: fine-tuning the pre-trained model to unlearn a specific concept while either constraining weight updates or replaying the remaining dataset to prevent degradation. In contrast, our method takes a fundamentally different approach. Instead of fine-tuning, we train a single Sparse Autoencoder (SAE) on the activations of the pre-trained model. We then identify sparse features associated with the unwanted concepts and block them to prevent their generation. To our knowledge, this is the first method to introduce such an approach.
>
> Our distinctive idea brings several benefits compared to recent approaches:
> - We demonstrate its effectiveness in unlearning styles and objects and further validate its practical utility by applying it to nudity unlearning (see Table 2 of additional results).
> - Due to the inherent nature of SAEs, the features selected for unlearning are highly interpretable (see Figures 3–8), enhancing the transparency of our approach.
> - Because our method blocks sparse and highly selective features, intervening on activations of a model, it is robust against adversarial attacks and can be used to unlearn multiple concepts at the same time (up to 49 out of 50 available styles, as presented in Table 4).
>
> Those properties distinguish SAeUron from existing solutions in terms of capabilities.
>
> >When talking about unlearning, a key issue is whether the erasing affects other output, especially similar concepts or objects. The paper has mentioned retention accuracy in the appendix but did not clearly state the evaluation criteria and the meaning of the results. Please elaborate it.
>
> We fully agree that retaining the remaining concepts is crucial for any unlearning method. This is why, in our studies, we employ the UnlearnCanvas benchmark designed to specifically measure not only the effectiveness of unlearning but also the retention accuracy. In particular, in Table 1, presenting the main results for different unlearning methods, we include the *In-domain retain accuracy (IRA)* that measures the quality of generation of other concepts when unlearning a particular one (e.g., 49 other styles when unlearning a single one), and *cross-domain retain accuracy (CRA)* which assesses the retention quality in a different domain (e.g., in style unlearning, we calculate object classification accuracy). As visible, our approach achieves state-of-the-art performance on those metrics in style unlearning and very high results for the object unlearning.
> To further strengthen our analysis of retention accuracy during unlearning, Fig 9 of additional results presents accuracy on each of the 20 classes from the UnlearnCanvas benchmark during unlearning. In general, our method successfully removes targeted classes while preserving the accuracy of the remaining ones. Nonetheless, in extreme cases where classes are highly similar to each other (e.g., Dogs and Cats), removing one of them negatively impacts the quality of the other. This observation is consistent across evaluated methods as presented in appendix D.3 of the original [UnlearningCanvas benchmark](https://arxiv.org/abs/2402.11846).
>
> To further evaluate the retention capabilities of our approach, **we run an additional experiment with an extreme scenario where we unlearn 49 out of 50 styles present in the UnlearnCanvas benchmark** (leaving one style out to evaluate the quality of its preservation). We observe almost no degradation in SAeUron performance for three randomly selected combinations. **For unlearning of 49/50 styles simultaneously, we observe UA: 99.29%, IRA: 96.67%, and CRA: 95.00%**. Those results highlight the unprecedented degree of unlearning precision with SAeUron.
>
> We are happy to discuss any other issues or questions regarding our paper. If there are none, we would highly appreciate you considering raising the score.

---

### Decision · Program_Chairs · 2025-05-01

**Decision:**

Accept (poster)

**Comment:**

The concerns raised by the reviewers were addressed in the rebuttal. Three out of the four reviewers have updated their recommendations to accept the paper. The remaining issues raised by the one reviewer who  recommended weak rejection have been checked by the AC. The AC thinks the responsed would have convinced the reviewer. As a result, the AC assumes that this reviewer will increase their score. In summary, this paper would have received a positive rating, and the AC sees no reason to overturn the consensus of the reviewers. Therefore, the AC recommends accepting the paper.